# SUBSTITUTING FROM THE INPUT: DISTILLING SEQUENTIAL COMPUTATION IN TRANSFORMER LANGUAGE MODELS

## ABSTRACT

Transformer language models process input sequences token by token, resulting in significant computation even when adjacent tokens are semantically redundant or compressible. We introduce a method for distilling sequential computation by replacing spans of input tokens with collapsed representations, computed on the fly by a shared, lightweight merge module. This module generates a single surrogate embedding from static token embeddings that captures the functional role of multiple tokens—without relying on model internals or context—allowing pre-trained models to operate on compressed inputs without architectural changes or re-training. We apply this approach during inference to compress both prompts and intermediate decoding steps, using a rollback mechanism to substitute stored multi-token KV cache entries with their single-step surrogates. Experiments with GPT-2 XL, LLaMA 3.1 8B, LLaMA 3.2 1.5B, and DeepScaleR across language modeling and downstream tasks (question answering, summarization, math reasoning) show up to 40% reduction in effective sequence length, with minimal accuracy degradation. These results highlight that sequential token computation in Transformers can be effectively approximated through condensed surrogate representations that preserves functional input behavior without model updating.

## 1 INTRODUCTION

Large language models (LLMs) have achieved remarkable success across a wide range of language understanding and generation tasks. Typically built on the Transformer architecture (Vaswani et al., 2017), these models process input sequences token by token, resulting in substantial computational overhead—both in terms of storing sequential states and executing autoregressive decoding. For instance, the phrase "My Neighbor Totoro" is split into four tokens by GPT-4o,[1] even though its meaning is clearer when treated as a whole. This prompts the question: can a Transformer's sequential computation be compressed mid-context—replacing multiple tokens with a single representation—even if the model is pre-trained with fixed tokenization and stepwise processing?

Prior work has explored reducing the sequential computation of Transformer language models at various levels. One line of research modifies the tokenization schemes on which models are trained, ranging from byte streams (Pagnoni et al., 2024) or patches (Xue et al., 2022; Yu et al., 2023) to "superword" phrases (Liu et al., 2025), but they require costly *re-training* of LMs, since tokenization defines the input structure before model construction. Another direction targets *pre-trained models*, aiming to execute multi-token computation with a single pass. Techniques like Copy-Generator (CoG) (Lan et al., 2023), speculative decoding (Leviathan et al., 2023), and CD-LM (Li et al., 2025) achieve this using additional components such as extended vocabularies, drafting models, or external chunk databases, but with substantial overhead.

In parallel, work on context reduction focuses on pruning or compressing inputs or intermediate model states (for example, the KV cache). While some of these methods operate post-training, they often depend on internal model signals—such as attention maps (Zhang et al., 2023) or hidden states (Bolya et al., 2023)—or require model re-training with custom compression objectives (Bolya et al.,

---

[1] https://platform.openai.com/tokenizer.

2023; Mu et al., 2023; Kallini et al., 2024). Moreover, techniques designed for long-context efficiency typically trade off generation accuracy (Jiang et al., 2023; Xiao et al., 2024; Shao et al., 2024). It remains unclear whether multi-step Transformer computations can be replaced with single-step surrogates post-training, without access to model internals or re-training, and without sacrificing the structural inductive biases of token-based generation.

In this work, we address the above question by distilling the sequential computation of multiple tokens into a single-step input representation for pre-trained language models. Given any pre-trained LM and a span of input tokens in context during generation, we aim to construct a *single surrogate embedding* that replaces the original token span *at the input level*, while preserving the model's next-token predictive distribution. To efficiently handle the combinatorial number of possible token spans during inference, we introduce a *shared, lightweight merge module* that amortizes the computation of surrogate embeddings. In contrast to prior approaches that depend on the language model's internal signals or surrounding context (Zhang et al., 2023; Mu et al., 2023; Ge et al., 2024; Shao et al., 2024), our merge module operates *solely on static input token embeddings*, producing a compressed representation that is agnostic to the context, token span length, and token identities. We refer to this method as **U**niversal **M**ulti-step **I**nput **M**erging (UMIM).

We implement the UMIM module as a lightweight, single-layer attention network that serves as a plug-in to any pre-trained Transformer language model, applied to input embeddings *prior to model computation* at inference. UMIM is trained using a predictive distillation loss, encouraging the model's output distribution with merged input embeddings to match that of the original uncompressed token sequences—thereby preserving generation behavior. Crucially, the pre-trained language model remains *frozen and black-box*, with only the external merge module updated during training. To construct training data, we extract frequent n-gram spans from a text corpus as candidate merge segments. Once trained, the same UMIM module can be applied across different downstream tasks without re-training, offering a general and reusable mechanism for input-level sequential computation compression.

Using the shared UMIM module over sequential input token embeddings, we compress generation contexts both in static prompts and dynamically during autoregressive decoding. At each step, a rollback mechanism replaces previously stored multi-token KV cache entries with their single-step surrogate, resulting in reduced sequence lengths and smaller KV cache footprints—thereby improving inference efficiency. Experiments across multiple pre-trained language models, including GPT-2 XL, LLaMA 3.1 8B, LLaMA 3.2 1.5B(evaluated on the Wikipedia corpus), and DeepScaleR (on advanced reasoning data), validate that UMIM preserves next-token predictive distributions after input merging. Extensive evaluations on downstream tasks—such as language modeling, question answering, summarization, and mathematical reasoning—demonstrate effectiveness of our approach, where UMIM reduces effective sequence lengths by up to 40%, with minimal degradation in performance compared to competitive baselines.

## 2 RELATED WORK

**Tokenization Variations and Extension** Several works explore alternative tokenization strategies for Transformer language models. Byte-level and tokenizer-free models such as ByT5 (Xue et al., 2022), Megabyte (Yu et al., 2023), and ByteFusion (Pagnoni et al., 2024) aim to reduce reliance on predefined vocabularies. SuperBPE (Liu et al., 2025) introduces "superword" tokens—phrase-level units including whitespace—to reduce sequence length and improve efficiency.

Other approaches extend token representations to support multi-token decoding. CoG (Lan et al., 2023) augments the vocabulary with span-level embeddings, enabling a "copy-and-paste" generation paradigm, though it requires retraining to support the extended space. CD-LM (Li et al., 2025) retrieves contextualized chunk embeddings from an external vector database for chunk-level decoding, interleaved with standard token generation, and operates without retraining.

Most of these methods require modifying the tokenizer or retraining the model, limiting their applicability to pre-trained LMs. In contrast, our work seeks to construct equivalent super-token embeddings post-training, without modifying the model or relying on external resources such as vector databases, enabling efficient input compression while preserving model behavior.

**KV Cache Management**   To reduce memory usage and improve inference efficiency during autoregressive decoding, recent work explores KV cache management strategies, which can be broadly categorized as static or dynamic. Static methods filter tokens during the prefill stage and retain them throughout decoding. In contrast, dynamic methods update the KV cache during generation, selectively retaining or evicting tokens. For example, $H_2O$ (Zhang et al., 2023) maintains a fixed-size cache by continuously pruning less important tokens. StreamingLLM (Xiao et al., 2024) leverages "attention sink" tokens and a sliding window over recent inputs to enable streaming inference over long contexts. KeyFormer (Adnan et al., 2024) improves this by dynamically selecting tokens based on their softmax contribution. OmniKV (Hao et al., 2025) prioritizes tokens using inter-layer attention similarity, offloading less critical KV pairs to CPU and reloading them on demand, avoiding permanent deletion. Other approaches, such as PagedAttention (Kwon et al., 2023), adopt hierarchical memory strategies, while Blockwise Caching (Dao et al., 2022) improves retrieval efficiency by operating at the block level rather than per-token. We reduce KV cache on the input level by replacing multi-token span KV cache with single step KV states, derived from our universal merge module for surrogate input embeddings.

**Context Compression, Token Pruning and Merging**   Reducing inference-time computation by compressing input contexts has been explored through both task-aware and task-agnostic approaches. Task-aware methods such as LongLLMLingua (Jiang et al., 2024) and Recomp (Xu et al., 2024) tailor compression based on downstream tasks, but typically require task-specific supervision and fixed compression ratios, limiting generality. Task-agnostic methods aim for broader applicability. LLMLingua (Jiang et al., 2023), LLMLingua2, and self-information filtering (Li, 2023) compress prompts using learned token salience or entropy-based scores. Other approaches like Gist Tokens (Mu et al., 2023), ICAE (Ge et al., 2024), and Extensible Tokenization (Shao et al., 2024) compress long contexts into vector representations, though often require training or modifying the LM itself.

Additional work explores context reduction through token pruning or merging. MRT5 (Kallini et al., 2024) discards low-importance tokens early in the encoder via a learned delete-gate. Merging techniques based on hidden states have been used in language (Yuan et al., 2024), vision (Bolya et al., 2023; Shin et al., 2025), video (Ryoo et al., 2021), and robotics models (Han), but generally depend on model internals or re-training.

In contrast, our approach is model-agnostic and operates externally, requiring no access to internal states or re-training. The UMIM module offers near-lossless span-level input compression inside the context through a lightweight merging mechanism applied directly to input embeddings, reducing both sequence length and KV cache usage at inference time, and integrating seamlessly with any pre-trained Transformer.

## 3   METHODOLOGY

Our core idea is to develop a Universal Multi-step Input Merging module, or UMIM, that acts as an external plug-in to pre-trained Transformer language models. The module compresses multiple adjacent input token embeddings into a *single surrogate embedding*, which drives the language model to produce equivalent next-token predictive distributions—effectively distilling multi-step computation into a single step.

The UMIM module operates solely on static input embeddings, independent of surrounding context. This enables it to function as a *universal replacement* for any repeated span of contiguous tokens, preserving their functional role across different contexts. It is designed to work post hoc with frozen language models, requiring no access to internal states or re-training.

Formally, let $\mathbf{x} = (x_1, \ldots, x_T)$ be a sequence of input tokens, and $e(\cdot)$ denote the token embedding function that maps each token to a $d$-dimensional vector. A Transformer language model with parameters $\theta$ (excluding the embedding layer) processes the embedded prefix $e(x_{<t}) = (e(x_1), \ldots, e(x_{t-1}))$ and outputs the next-token distribution $p_\theta^t(\cdot \mid e(x_{<t}))$ at *token step* $t$. We aim to learn a general merge module $\mathcal{M}_\phi$, parameterized by $\phi$, that operates on a span of $n$ contiguous tokens $x_{t:t+n} = (x_t, \ldots, x_{t+n-1})$ and produces a surrogate embedding $\mathcal{M}_\phi(e(x_{t:t+n}))$ to replace the original span of embeddings $e(x_{t:t+n}) = (e(x_t), \ldots, e(x_{t+n-1}))$, without altering the subsequent predictive distribution from the language model.

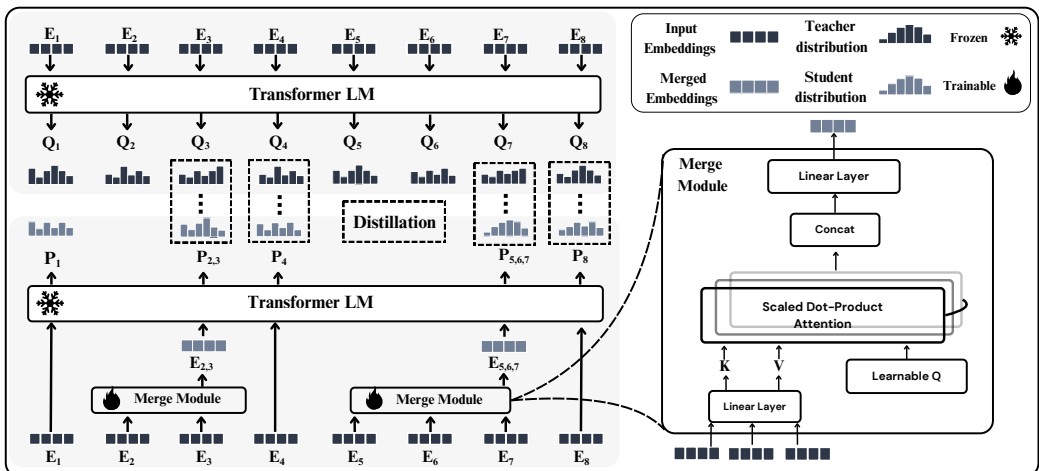

Figure 1: Training process of the UMIM module, with token-step aligned distillation loss and merge module architecture.

In the following sections, we describe a simple model-agnostic heuristic for selecting token spans to merge (Sec.3.1), the architecture of the lightweight merge module (Sec.3.2), the training objective for learning it via a sequential computation distillation loss (Sec.3.3), and decoding algorithms that apply UMIM during both prompt compression and runtime generation (Sec.3.4) to improve efficiency and scalability.

## 3.1 MERGE RULE

To determine which token spans to merge, we adopt a simple, model-agnostic heuristic based on n-gram frequency. Given a general text corpus $\mathcal{C}$, we extract a set $\mathcal{R}$ of high-frequency n-grams—contiguous token spans—that are eligible for merging. Specifically, we collect all unique n-grams (up to a maximum length $n = 4$) whose frequency exceeds a threshold $\tau$, forming the *merging set* $\mathcal{R}$. This cap on $n$ mitigates data sparsity while allowing for meaningful multi-token groupings.

In cases where nested n-grams appear (e.g., overlapping candidates), we apply a longest-match strategy, giving precedence to the longest valid span. During both training of the merge module $\mathcal{M}_\phi$ and inference-time compression, all token spans belonging to $\mathcal{R}$ are considered merge candidates.

Importantly, the merge rule is independent of the model architecture and depends only on the tokenizer and corpus statistics. It is worth noting that the corpus $\mathcal{C}$ used to construct $\mathcal{R}$ is task-independent, but using in-domain data may encourage more effective compression by better capturing frequently co-occurring token spans. Se details in Appendix A.

## 3.2 LIGHTWEIGHT MERGE MODULE ARCHITECTURE

We design the UMIM merge module as a lightweight attention-based neural network that operates solely on static input embeddings, prior to any Transformer computation. Its purpose is to serve as a universal replacement for repeated spans of contiguous tokens across different contexts, fully capturing their functional role with a single surrogate embedding. Importantly, the module supports variable-length token spans, handled flexibly by its attention mechanism.

The merge module is implemented as a single-layer multi-head attention network, following the structure of the original Transformer (Vaswani et al., 2017), but with a *learnable query*. This allows the module to selectively aggregate span representations into a fixed-size output. Formally, let $d$ denote the embedding dimension of the target language model, and $h$ the number of attention heads. Each head has dimension $d_h = d/h$. Given the token span embeddings as $e(x_{t:t+n}) \in \mathbb{R}^{d \times n}$.

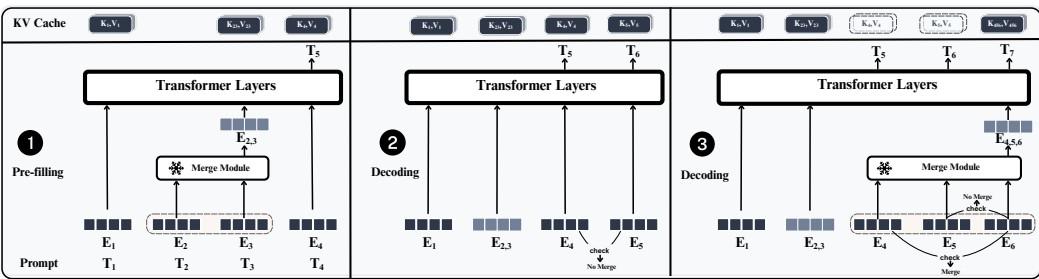

Figure 2: UMIM Decoding Process

The learnable query matrix is $Q = [q_1; \ldots; q_h] \in \mathbb{R}^{d \times 1}$, composed of $h$ individual query vectors $q_i \in \mathbb{R}^{d_h \times 1}$. The output surrogate embedding is computed as:

$$\mathcal{M}_\phi \left( e(x_{t:t+n}) \right) = W^o [o_i; \cdots; o_h]^\mathsf{T}; \quad o_i = \mathrm{softmax} \left( \frac{q_i^\mathsf{T} \left( W_i^k e(x_{t:t+n}) \right)}{\sqrt{d_h}} \right) (W_i^v e(x_{t:t+n}))^\mathsf{T}$$

where the merge module learnable parameters are $\phi = \{Q, \{W_i^k\}_{i=1}^h, \{W_i^v\}_{i=1}^h, W^o\}$ and $W_i^k, W_i^v \in \mathbb{R}^{d_h \times d}, W^o \in \mathbb{R}^{d \times d}$. The total number of parameters is $d + 3d^2$.

## 3.3 UMIM Learning by Distilling Sequential computation

We train the UMIM merge module to produce single-step surrogate embeddings that can replace multi-token input sequences, while preserving the language model's next-token predictive distributions. The language model parameters $\theta$ are frozen throughout training and treated as a black box. To guide learning, we introduce a *sequential computation distillation loss*, aligning predictions made on merged inputs with those from the original token-by-token computation.

Let $\mathcal{M}_\phi(e(x_{1:T})) = (e'_s)_{s \in \mathcal{S}}$ denote the compressed sequence of token embeddings after applying the merge module to all eligible token spans according to the merge rule $\mathcal{R}$. Here, $\mathcal{S}$ represents the set of *retained token positions* after merging. For instance, if tokens $x_{t:t+n}$ are merged into a single surrogate embedding $\mathcal{M}_\phi(e(x_{t:t+n}))$, then token steps $t$ through $t + n - 1$ are collapsed, and only step $t + n$ is retained in $\mathcal{S}$ (e.g., $\mathcal{S} = (\cdots, t - 1, t + n, \cdots)$).

To align the behavior of the compressed input with the original, we minimize the KL divergence between the output distributions of the language model on unmerged vs. merged input prefixes:

$$\mathcal{L}(\phi) = \sum_{s \in \mathcal{S}} D_{\mathrm{KL}} \left( p_\theta(\cdot | e(x_{<s})) \, \| \, p_\theta(\cdot | \mathcal{M}_\phi(e(x_{<s}))) \right) = - \sum_{s \in \mathcal{S}} p_\theta(\cdot | e(x_{<s})) \log p_\theta(\cdot | \mathcal{M}_\phi(e(x_{<s})))$$

This loss encourages the merged representation to produce predictive distributions that match those from the full token sequence. It is equivalent to a cross-entropy loss with a soft teacher distribution from the uncompressed model. An illustration of the training procedure is shown in Figure 1.

Only the parameters $\phi$ of the merge module are updated during training; the language model remains unchanged. This setup enables us to distill multi-step computation into a single embedding step, using amortized merging over static input embeddings. Crucially, the merge module learns to represent each frequent $n$-gram with a surrogate embedding that captures its **semantic role** (see Appendix F.1.2, Figure 3) *in arbitrary contexts of the backbone model*. In practice, the surrogate embedding replaces the original token-level embeddings of the span in the input, serving as a drop-in substitute that preserves the predictive distribution of the language model. Because the loss explicitly enforces alignment of merged and unmerged prefixes, UMIM does not alter contextual semantics but instead compresses them into a more compact representation.

## 3.4 Model Inference with Runtime Merging

Once trained, the UMIM module can be integrated into inference with the compatible pre-trained language model. By replacing spans of input tokens with their corresponding surrogate embeddings,

UMIM reduces the effective sequence length, leading to smaller KV caches and more efficient autoregressive decoding.

Let $z = (z_1, \cdots, z_m)$ denote the prompt tokens, and $y = (y_1, y_2, \cdots)$ the tokens generated by the model. The UMIM module can be applied to both the static prompt and the dynamic generation process. An illustration is in Figure 2.

For the prefilling, we scan the token sequence for eligible n-grams defined in the merging set $\mathcal{R}$. Each matched span is replaced by its surrogate embedding from UMIM, resulting in a compressed input embedding sequence passed to the language model.

During decoding, after each new token is generated, we check whether the most recent n-gram (ending in the new token) belongs to $\mathcal{R}$. If it does, we **roll back** the last $n$ steps by removing the corresponding KV cache entries and replacing them with a single surrogate embedding. This surrogate is then re-encoded and reinserted into the cache, effectively merging the span and continuing generation with reduced sequence length.

Since UMIM operates by directly replacing $n$-gram spans in the input or decoding sequence with single surrogate embeddings, it does not introduce additional inference overhead beyond lightweight span checks. On the contrary, the resulting reduction in sequence length directly reduces the size of the memory and attention computation. A detailed efficiency analysis is provided in Appendix F.3.

## 4 EXPERIMENTS

### 4.1 EXPERIMENTAL SETUP

We train UMIM modules for a variety of pre-trained LLMs of different sizes, including Llama 3.1 8B Grattafiori et al. (2024),Llama 3.2 1.5B, GPT2-XL 1.5B Radford et al. (2019) using the WikiText-103 (Merity et al., 2016) corpus, and DeepScaleR-1.5B-Preview[2] using a long CoT (chain-of-thought) reasoning trajectory corpus we collected. We also trained on larger context(on the Wikipedia and Bookcorpus)) on Gemma3 4B Team et al. (2025) as abalation study(See Appendix G) For the merge module, we have number of attention heads $h = 4$, embedding size $d = 4096, 2048, 1600, 1536$ to be compatible with the above models, respectively. This results in learnable parameter size in the merge modules from 50M to 7M, which is very light. To build the merging rule $\mathcal{R}$, we extract $n \in \{2, 3, 4\}$-grams with the same frequency threshold $\tau = 5$. The resulted merging set covers over 2 million unique token spans for each model. More details can be found in Appendix A

We first measure the distribution alignment quality after applying the embedding merges compared with the original token-based LLMs. The metrics include: Top-1 accuracy—percentage of token positions after merging where the top-1 tokens in the predictive distribution exactly match, Top-3/10 overlap—average number of overlapped tokens in the top-3/10 tokens from corresponding distributions at aligned token positions, Top-p overlap—similar to top-3/10 but the number of tokens in each distribution to look at is decided by the top-p accumulative probability mass, and Mean Reciprocal Rank (MRR)—the average reciprocal rank of the teacher (original LLM)'s top-1 token within the distribution after span merging at corresponding token positions.

Further, we measure the language modeling performance using perplexity (PPL) with UMIM on different test datasets (Sec. 4.3), including WikiText-103 where the module is trained on, as well as BookCorpus (Zhu et al., 2015) and OpenWebText (Gokaslan & Cohen, 2019) with the same merge module without re-training. To measure efficiency gain with span merging, we measure the percentage of token reduction, or TR (%), directly reflecting the saved sequence length and KV cache.

Finally, we assess various downstream task performances with UMIM applied on **question answering (QA)**, **document summarization**, and **math reasoning** to test **long context** generation ability (Sec. 4.4). Note that UMIM is trained *once* on WikiText-103 only, without any task-specific supervision. The same trained module is then directly applied to all downstream tasks, without further fine-tuning, highlighting the **task-agnostic** nature of our approach. We compare with context reduction baselines including Select Context (Li et al., 2023), Llmlingua1 (Jiang et al., 2023), Llmlingual2 (Pan et al., 2024), H2O(Zhang et al., 2023) , and StreamingLLM (Xiao et al., 2024), which are representative approaches that are most relevant to this line of work.

---

[2]`https://huggingface.co/agentica-org/DeepScaleR-1.5B-Preview`.

Ablation study on different components of our approach is presented in Appendix G.

## 4.2 PREDICTIVE DISTRIBUTION ALIGNMENT

Table 17 shows results of our merge module evaluated on the corresponding *test* split after training. The higher the values are, the better the distributions after applying token merging are matched with the original distribution at aligned token positions. We observe a high matching scores for the distribution, validating the effectivenes of the universal token merging module even on static input embeddings in context. Furthermore, we achieve high token reductions between 30-60%, leading to great potentials for efficient inference.

Table 1: Predictive distribution alignment metrics, with token reduction (TR) rate.

| Model | TR (%) | Top-1 | Top-3 | Top-10 | Top-$p$ | MRR |
|---|---|---|---|---|---|---|
| Llama-3.1-8B | 37.8% | 79.22 | 77.56 | 77.58 | 92.91 | 87.05 |
| Llama-3.2-1.5B | 37.8% | 74.10 | 74.14 | 75.28 | 91.55 | 83.30 |
| GPT2-XL | 35.74% | 69.55 | 71.28 | 73.59 | 90.65 | 79.28 |
| DeepScaleR-1.5B-Preview | 54% | 77.00 | 69.30 | 67.99 | 90.98 | 85.59 |

## 4.3 LANGUAGE MODELING AND GENERATION

We present language modeling evaluation measured by PPL in Table 2 across three different test data, along with the token reduction rate. All merge modules are only trained on WikiText-103 training set, and merge rules are derived from there too.

We show that UMIM works well when generalizing to unseen data (BookCorpus and OpenWebText) without re-training, evidenced by the PPL values close to the original token-based processing. One exception is the GPT2-XL model, where it demonstrates much worse PPL compared to the base model when combined with token merging. We hypothesize that this is primarily due to GPT2-XL's comparatively limited intrinsic generation capability, compounded by its merge module containing fewer parameters—a hypothesis substantiated by training results.

In terms of token reduction, unsurprisingly the WikiText test split, belong to the same domain as the merge module training, exhibits a higher number of merges with close to 40% reduced tokens compared to the other two testsets from distinct domains. Nonetheless, 12-15% token reduction is still observed in generalized domains. We hypothesize that at the scale of learning data grows, the generalization in terms of efficiency will increase. In summary, UMIM exhibits strong and robust capabilities across various pretrained LMs, even on data outside of the training domain, highlighting its generalizability and versatility.

We also show qualitative generation results in Table 3, using greedy decoding from the Llama 3.1 8B model. We observe that in some cases, token merging does not alter the original output of the model. In other cases, although the outputs differ slightly from the original ones, the generated content remains coherent and semantically reasonable. These results indicate that incorporating our merge module enables effective token compression, reducing the number of tokens while preserving output fidelity. In certain instances, the merge module can even achieve lossless compression of tokens without any impact on the generated output(See more results in Appendix F).

## 4.4 DOWNSTREAM TASKS

**QA and Summarization**  With the **same** merge modules trained in WikiText-103, we directly apply them to downstream QA and summarization tasks. We test on PIQA (Bisk et al., 2020), Copa (Gordon et al., 2012), and OpenBookQA (Mihaylov et al., 2018), ARC_Easy, ARC_Challenge (Clark et al., 2018), with reported zero-shot QA accuracy in Table 4. For summarization we test on CNN/DailyMail (Nallapati et al., 2016), with metrics including Rouge (Ganesan, 2018) and BERTScore (Zhang et al., 2020), reported in Table 5. More implementation details can be found in Appendix F.1. Efficiency metric of token reduction (TR) is controlled for all baselines to be the same as our approach for fair comparison.

Table 2: Language modeling performance (PPL) and token reduction (TR).

| Methods | WikiText-103 | | BookCorpus | | OpenWebText | | PPL Avg. |
|---|---|---|---|---|---|---|---|
| | PPL | TR (%) | PPL | TR (%) | PPL | TR (%) | |
| Llama-3-8B | 13.386 | – | 15.255 | – | 9.220 | – | 12.620 |
| Merge Module | 13.814 | 36.09% | 15.258 | 14.78% | 9.818 | 13.76% | 12.963 |
| Llama-3.2-1B | 20.050 | – | 21.189 | – | 13.369 | – | 18.869 |
| Merge Module | 23.439 | 36.09% | 21.954 | 14.78% | 14.740 | 13.76% | 20.711 |
| GPT2-XL | 28.555 | – | 27.095 | – | 13.050 | – | 22.233 |
| Merge Module | 37.860 | 35.10% | 29.970 | 14.09% | 14.392 | 12.45% | 27.407 |

Table 3: Qualitative generation results. Highlighted texts are merged, with [] for merging boundaries.

| Prompt | Generation | |
|---|---|---|
| | Base Model | Merge Module |
| Traditional Chinese literary criticism emphasized **[the life of the]** author when interpreting a work | The author's life was seen as the key to understanding the work. | The author's life **[was seen as]** the key **[to understanding the work]**. |
| In **[2025]** , **[he moved to the]** institute **[in an attempt]** to study. | He was a very good student, and he was very good at his job. | He **[was a student of]** **[the institute and]** **[was a member of]** the institute's football team. |
| In the spring , **[his youngest son]**, Ian, was born. | He was a beautiful baby boy, and he was the light of his father's life. | He **[was a beautiful baby]**, **[with a head of thick]**, dark **[hair and]** a round, chubby face. |

On both tasks, UMIM shows superior performance compared to all baselines at the same token reduction rate, even in domains and tasks entirely unseen during training. This confirms not only the strong capability of our approach in maintaining model generation capabilities, but also highlights its generalization across downstream tasks without re-training.

Table 4: QA task performances, along with token reduction (TR).

| Methods | PIQA | | Copa | | OpenBookQA | | ARC-Easy | | ARC-challenge | | Avg (%) |
|---|---|---|---|---|---|---|---|---|---|---|---|
| | Acc.% | TR% | Acc.% | TR% | Acc.% | TR% | Acc.% | TR% | Acc.% | TR% | |
| Llama-3.1-8B | 79.60 | – | 76.80 | – | 43.60 | – | 76.49 | – | 49.50 | – | 65.19 |
| Merge Module | **78.40** | 9.68 | **76.80** | 5.90 | **42.40** | 9.77 | **73.68** | 13.76 | **44.82** | 14.18 | **63.22** |
| Select Context | 73.34 | 10.00 | 72.60 | 6.00 | 40.80 | 10.00 | 57.2 | 14.00 | 43.48 | 10.00 | 53.48 |
| Llmlingua2 | 75.14 | 10.00 | 72.20 | 6.00 | 32.80 | 10.00 | 44.04 | 14.00 | 32.78 | 14.00 | 51.39 |
| Llama-3.2-1B | 75.52 | – | 70.60 | – | 36.00 | – | 58.77 | – | 35.45 | – | 56.26 |
| Merge Module | **72.20** | 9.68 | 68.80 | 5.90 | 31.20 | 9.77 | **55.26** | 13.76 | **33.44** | 14.18 | **52.3** |
| Select Context | 68.82 | 10.00 | 65.80 | 6.00 | **35.00** | 10.00 | 46.49 | 14.00 | 32.78 | 14.00 | 49.71 |
| Llmlingua2 | 69.10 | 10.00 | **69.20** | 6.00 | 31.20 | 10.00 | 37.89 | 14.00 | 26.42 | 14.00 | 46.76 |
| GPT2-XL | 70.29 | – | 65.20 | – | 28.00 | – | 47.72 | – | 29.77 | – | 48.18 |
| Merge Module | **68.82** | 9.08 | 62.00 | 9.12 | **29.00** | 9.70 | **43.33** | 13.82 | **26.76** | 14.14 | **45.98** |
| Select Context | 64.80 | 9.00 | 61.00 | 9.00 | **29.00** | 10.00 | 40.88 | 14.00 | 25.42 | 14.00 | 44.22 |
| Llmlingua2 | 66.59 | 9.00 | **63.80** | 9.00 | 26.40 | 10.00 | 30.70 | 14.00 | 25.75 | 14.00 | 42.64 |

[1] We evaluated the effectiveness of our trained merge module on QA tasks, using accuracy as the primary metric (higher is better).

Table 5: Task of CNN/DailyMail summarization performance with controlled token reduction (TR).

| Methods | TR (%) | Rouge-1 | Rouge-2 | Rouge-L | Rouge-Lsum | Avg | BERTScore |
|---|---|---|---|---|---|---|---|
| Llama-3.1-8B | 0% | 37.44 | 15.56 | 24.31 | 31.29 | 27.18 | 86.76 |
| Merge Module | 16% | **35.54** | **13.86** | **23.06** | **29.69** | **25.54** | **86.44** |
| Select Context | 16% | 33.53 | 11.20 | 21.43 | 27.55 | 23.43 | 85.08 |
| Llmlingua1 | 16% | 33.55 | 10.31 | 21.33 | 27.66 | 23.21 | 85.33 |
| Llmlingua2 | 16% | 33.04 | 11.32 | 22.98 | 28.75 | 24.02 | 84.96 |
| StreamingLLM | 16% | 28.81 | 10.06 | 18.91 | 24.29 | 20.52 | 84.47 |
| H2O | 16% | 24.42 | 9.28 | 16.29 | 21.24 | 17.81 | 82.73 |

**Advanced Math Reasoning** We conduct evaluations on the AIME 2024 (Mathematical Association of America, 2024) and AMC mathematics benchmarks using the DeepScaleR model to test long context generation reasoning ablility, with UMIM trained on collected long CoT reasoning trajectories. We set the maximum generation length to 15K tokens and assess performance using the pass@k accuracy metric. Experimental results indicate that our merge module consistently achieves strong performance on these challenging reasoning tasks. Notably, our method outperforms both Select Context and LLMLingua baselines, further underscoring its effectiveness and generalizability in reasoning-intensive applications with long CoT.

Table 6: Advanced reasoning task performance with token merging.

| Methods | AIME 2024 | | AMC 2023 | |
|---|---|---|---|---|
| | Accuracy | TR (%) | Accuracy | TR (%) |
| DeepScaleR | 15/30 | – | 37/40 | – |
| Merge Module | **13/30** | 40% | **32/40** | 42.40% |
| Select Context | 8/30 | 40% | 21/40 | 42.40% |
| Llmlingua2 | 11/30 | 40% | 28/40 | 42.40% |

## 4.5 ABLATION STUDY

We ablate components of the UMIM pipeline, including the construction of the merging set (n-gram frequency threshold and span extraction), parameterization of the merge module (layers and size), and decoding strategies.

We observe that smaller frequency thresholds $\tau$ degrade performance by causing excessive merges. The merge module's effectiveness depends not only on parameter capacity—though it remains lightweight—but also on the volume of training data, which improves generalization. Full results are in Appendix G.

## 5 CONCLUSION

We introduce a novel framework for compressing sequential computation in Transformer language models through a lightweight, universal input merging module, UMIM. By replacing spans of adjacent tokens with a single surrogate embedding at the input level, UMIM enables both prompt and runtime compression without modifying the underlying model architecture or requiring access to model internals. Trained via a distillation objective that aligns next-token distributions, the merge module captures the functional role of multi-token sequences in a compact form.

Our approach is **model-agnostic**, requires no re-training of the language model, and operates solely on static token embeddings. At inference, UMIM reduces effective sequence lengths and KV cache usage by dynamically merging generated tokens on the fly, improving memory and computational efficiency. Experiments across diverse models and tasks demonstrate that this strategy achieves minimal performance degradation.

LIMITATIONS

While UMIM offers a flexible and efficient mechanism for distilling sequential computation, several limitations remain. First, although the language model remains frozen, the merge module still requires task-agnostic training using distillation, which introduces additional overhead and limits plug-and-play applicability.

Second, the current approach relies on a fixed merging set $\mathcal{R}$ derived from n-gram frequency, and does not adaptively control merge frequency or merge decisions at inference time. This lack of dynamic control may result in suboptimal trade-offs between compression and fidelity, especially in domains with high variability or long-tail phrasing.

Third, UMIM is not explicitly trained to handle novel or rare token spans outside the merging set, which may limit generalization. Moreover, while we demonstrate strong results across models up to the 8B scale, further evaluation is needed to assess effectiveness and scalability on larger models or in highly specialized domains.

Addressing these limitations—through adaptive span selection, controllable compression strategies, and broader training across diverse data distributions—remains a promising direction for future work.

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

# Technical Appendices

ORGANIZATION OF CONTENTS

## A    MERGING SET MINING DETAILS

Given a text corpus $\mathcal{C}$, we tokenize all documents using the base model's tokenizer. The resulting token sequence is partitioned into non-overlapping segments of length $L = 512/1024$ for efficient batch processing. In the following, we detail each step of our high-frequency $n$-gram mining .

**N-gram Extraction and Frequency Counting**    For each sequence, we enumerate all contiguous $n$-grams for $n \in \{2, 3, 4\}$ and compute their global frequency across the corpus. Let $S = \{s_1, s_2, \ldots, s_N\}$ denote the collection of tokenized segments, where $s_i = (t_{i,1}, t_{i,2}, \ldots, t_{i,L})$. For each $n$-gram $g = (t_j, t_{j+1}, \ldots, t_{j+n-1})$, we maintain a frequency count $f(g)$.

**Threshold-based Candidate Selection**    We retain only those $n$-grams whose corpus-wide frequency $f(g)$ exceeds a preset threshold $\tau$ (e.g., $\tau = 500, 50, 5$; see ablation study). This yields a high-frequency $n$-gram set $\mathcal{R} = \{g \mid f(g) \geq \tau\}$ for all $2 \leq n \leq 4$.

**Hierarchical Overlap Filtering**    Since $n$-grams may overlap (e.g., a trigram may share tokens with a bigram or fourgram), we apply a two-stage filtering procedure:

1. **Intra-set Filtering:** Within each $n$-gram set, we remove candidates that are substrings of longer $n$-grams with the same prefix or suffix, keeping only the highest-frequency candidates.

2. **Inter-set Filtering:** We further eliminate any $n$-gram that is fully contained within a higher-order $m$-gram ($m > n$) present in $\mathcal{R}$, so that only the longest, non-overlapping spans are retained as merge candidates.

**Reasons for 2- to 4-grams**    We restrict our merging set $\mathcal{R}$ to $n$-grams of length 2 to 4 for both linguistic and computational reasons. Empirically, 2- to 4-gram phrases correspond to the most meaningful word groups in natural language and align with common tokenizer outputs. Including much longer $n$-grams (e.g., 8-grams or 10-grams) would reduce the practical number of merges, as longer $n$-grams mask and exclude shorter, high-frequency candidates within their span. Additionally, for autoregressive generation, merging long $n$-grams is impractical since the model must wait until all tokens of a long span have been generated to determine if a merge is possible, introducing latency and reducing compression effectiveness. Using 2-4 grams, we achieve a balance between merge granularity, linguistic coverage, and computational efficiency.

**Different N-grams chunk numbers**    We present the specific merge rules used in our experiments, with the threshold $\tau$ set to 5 (see Section 3.1 and Section 4.1). For Llama-3.1-8B, Llama-3.2-1.5B, and GPT2-XL, the merge rules are derived from Wikitext-103. In contrast, the merge rules for DeepScaleR-1.5B-Preview are selected based on our own generated dataset (see Section E.1).

Table 7: Different N-grams chunk numbers

| Model | 2-grams | 3-grams | 4-grams |
|---|---|---|---|
| Llama-3.1-8B | 216700 | 684793 | 1266080 |
| Llama-3.2-1.5B | 216700 | 684793 | 1266080 |
| GPT2-XL | 218415 | 702570 | 1241657 |
| DeepScaleR-1.5B-Preview | 35691 | 212367 | 1665423 |

Pseudocode 1: N-grams Extraction

```
# input_ids  - [n, s], tokenized corpus (n seqs of length s)
# N          - list of n-gram lengths, e.g., [2,3,4]
# threshold  - minimum frequency

# extract all n-grams and count frequencies
ngram_freqs = {}
for n in N:
    grams = sliding_window(input_ids, n)        # [*, n]
```

```
    ngram_freqs[n] = count_unique(grams)          # {ngram: count}

# filter by frequency
high_freq = {n: {g for g,c in ngram_freqs[n].items() if c >= threshold}
    for n in N}

# remove n-grams that are subsumed by longer, high-frequency n-grams
for n in reversed(N):
    for g in set(high_freq[n]):
        if exists_longer_overlap(g, high_freq):
            high_freq[n].remove(g)

# final merge rule sets
bigrams, trigrams, fourgrams = high_freq[2], high_freq[3], high_freq[4]
```

## B MODULE ARCHITECTURE AND INTEGRATION

We describe more details of the merge module in Section 3.2 and how it is integrated into the base LM at input level.

We design the UMIM merge module as a lightweight, attention-based neural network that operates solely on static input embeddings, prior to any Transformer computation. Its primary function is to serve as a universal replacement for repeated spans of contiguous tokens across diverse contexts, producing a single surrogate embedding that captures the aggregate functional role of the original span. The module supports variable-length token spans (with $n$ ranging from 2 to 4 in our experiments), is flexibly handled by its multihead attention mechanism, and is agnostic to the underlying model architecture.

Formally, let $d$ denote the embedding dimension of the target language model and $h$ the number of attention heads, such that each head has dimension $d_h = d/h$. Given the token span embeddings $e(x_{t:t+n}) \in \mathbb{R}^{d \times n}$, the merge module operates as follows: a learnable query matrix $Q = [q_1; \ldots; q_h] \in \mathbb{R}^{d \times 1}$, composed of $h$ individual query vectors $q_i \in \mathbb{R}^{d_h \times 1}$, is used to attend over the span. For each head $i$, attention weights are computed via a scaled dot product between $q_i$ and a linearly projected version of the span embeddings ($W_i^k e(x_{t:t+n})$), resulting in normalized attention over the span. The output of each head is a weighted sum of linearly projected value vectors ($W_i^v e(x_{t:t+n})$). Finally, the outputs from all heads are concatenated and passed through an output projection $W^o$ to obtain the final surrogate embedding:

$$\mathcal{M}_\phi\left(e(x_{t:t+n})\right) = W^o[o_1; \cdots; o_h]^\intercal, \qquad o_i = \text{softmax}\left(\frac{q_i^\intercal W_i^k e(x_{t:t+n})}{\sqrt{d_h}}\right)(W_i^v e(x_{t:t+n}))^\intercal$$

where the learnable parameters are $\phi = \{Q, \{W_i^k\}_{i=1}^h, \{W_i^v\}_{i=1}^h, W^o\}$, with $W_i^k, W_i^v \in \mathbb{R}^{d_h \times d}$ and $W^o \in \mathbb{R}^{d \times d}$. The total number of parameters is $d + 3d^2$, which is lightweight relative to the base model.

During traning, the merge module is applied to all eligible spans as defined by the high-frequency $n$-gram merge rule $\mathcal{R}$. Each identified span is replaced by its corresponding surrogate embedding, and the resulting sequence is updated accordingly. The merge operation is performed once, prior to any downstream Transformer layers or autoregressive generation. This design ensures that the merge module can be integrated seamlessly into any Transformer-based language model pipeline, requires minimal overhead, and does not alter the base model parameters or architecture.

In summary, the UMIM merge module provides a flexible, learnable mechanism for compressing repeated multi-token spans into single embeddings, enabling both efficient inference and effective knowledge distillation without modifying the underlying language model structure.

Pseudocode 2: Multi-Head Pooling Module

```
# MultiHeadPooling module
# tokens          - (batch, k, d_model), input token representations
# num_heads       - number of pooling heads
# head_dim        - d_model // num_heads
```

```
# query           - (num_heads, head_dim), learnable pooling queries

# process tokens with token processor (MLP+GELU+LayerNorm)
tokens_proc = token_processor(tokens)              # (batch, k, d_model)

# reshape for multi-head
tokens_proc = tokens_proc.view(batch, k, num_heads, head_dim)
tokens_proc = tokens_proc.permute(0, 2, 1, 3)   # (batch, num_heads, k,
    head_dim)

# expand learnable query to batch
query = query.expand(batch, num_heads, 1, head_dim)

# compute attention scores (dot product + scaling)
attn_scores = einsum("bnhd,bnhd->bnh", query, tokens_proc) / sqrt(
    head_dim)

# softmax over k (tokens)
attn_weights = softmax(attn_scores, dim=2).unsqueeze(-1)   # (batch,
    num_heads, k, 1)

# weighted sum of tokens for each head
pooled = sum(tokens_proc * attn_weights, dim=2)            # (batch,
    num_heads, head_dim)

# flatten and project output
merged = pooled.view(batch, d_model)
output = output_proj(merged)                              # (batch,
    d_model)
```

## C  MULTI-STEP DISTILLATION LOSS AND ALIGNMENT METRICS

Here we describe in more detail the distillation loss from multi-step computation to compressed input after UMIM merging as introduced in Section 3.3, as well as how the learning algorithm is implemented, and how the loss aliment is evaluated with different metrics.

**Masking, Alignment, and Index Construction.**  Sequence length and target alignment are altered after token merging. To ensure correct supervision and evaluation, we precompute three masks and an index mapping for every input sequence:

- **Attention Mask**: A binary mask of shape $[\text{batch\_size}, \text{seq\_len}]$, where $1$ indicates valid tokens (including merged/retained tokens) and $0$ denotes padding. This mask is used for all Transformer operations.

- **Padding Mask**: A mask that distinguishes true tokens from padding after batch collation and zero-padding to a fixed length.

- **Loss Mask**: A binary mask marking tokens that should be counted toward the loss and metric computation. Importantly, this mask is constructed so that:
    - All tokens prior to the first merge are masked out ($0$).
    - Within any merged span, only the *last* (surrogate) token is marked as $1$; all others are set to $0$ to prevent repeated counting and label misalignment.
    - All padding tokens are masked out.

- **Index Mapping**: We store, for each sequence, the mapping from original to compressed token positions, so that teacher and student outputs can be aligned for supervision and metrics.

Before each forward pass and loss computation, we assert that the teacher and student masks are perfectly aligned; if not, an exception is raised to prevent silent misalignment.

**Loss Computation.**   We compute the distillation loss as the average cross-entropy between the predictive distributions of the base LLM (teacher) and the merge module (student), but crucially, the loss is only evaluated at positions directly influenced by the merge process. Specifically, we use a binary mask to select tokens for supervision:

- For all tokens before the first merge, the base LLM is frozen and the outputs of teacher and student are identical; these positions are excluded from loss computation as they provide no meaningful learning signal.

- For each merged span, only the final token is retained in the compressed sequence. The distillation target for each retained token is the original model's output at that position.

- The loss is normalized by the total number of supervised tokens (the sum of the mask), ensuring the reported value reflects only the positions where the merge module has influence.

Mathematically, for aligned and filtered outputs $p_i^{\text{teacher}}$, $p_i^{\text{student}}$ at valid positions $i$, the loss is:

$$\mathcal{L} = -\frac{1}{N_{\text{valid}}} \sum_{i=1}^{N_{\text{valid}}} \sum_v p_i^{\text{teacher}}(v) \log p_i^{\text{student}}(v)$$

where $N_{\text{valid}}$ is the total number of masked-in tokens (i.e., $\sum$ loss_mask over the batch).

**Cross-Entropy and KL Divergence:** Although distillation is often formulated with the Kullback–Leibler (KL) divergence, in practice, we use cross-entropy for computational efficiency and simplicity. The KL divergence between two distributions $P$ (teacher) and $Q$ (student) is:

$$D_{\text{KL}}(P\|Q) = \sum_v P(v) \log \frac{P(v)}{Q(v)} = H(P, Q) - H(P)$$

where $H(P, Q)$ is the cross-entropy and $H(P)$ is the (teacher) entropy. Crucially, $H(P)$ depends only on the (frozen) teacher and is constant with respect to the student/merge module parameters. Thus, minimizing $D_{\text{KL}}(P\|Q)$ is equivalent (in gradient and optimization) to minimizing the cross-entropy $H(P, Q)$. Therefore, we implement the distillation loss as cross-entropy for simplicity, omitting the constant $H(P)$ term.

This choice does not change the optimization or the learned solution, but avoids unnecessary computation and numerical instability from subtracting nearly equal terms.

**Metric Computation and Reporting.**   We assess model fidelity using several metrics, computed strictly on masked and aligned positions to ensure fair evaluation and to avoid score inflation or deflation due to padding or repeated tokens. The reported metrics are as follows:

- **Top-1 Accuracy**: At each valid position, we measure the proportion of cases where the token with the highest predicted probability (i.e., the greedy argmax) matches between the student and teacher distributions. This metric reflects exact prediction agreement.

- **Top-$k$ Overlap**: For $k = 3$ and $k = 10$, we compute the overlap ratio between the sets of top-$k$ predicted tokens from the teacher and student distributions at each valid position. The overlap is defined as the size of the intersection divided by $k$, with the final score averaged across all valid positions. This metric quantifies agreement within the highest-probability candidate sets.

- **Top-$p$ Overlap**: For each valid position, we identify the minimal sets of tokens in both the teacher and student distributions whose cumulative probability mass meets or exceeds a threshold $p$ (we report results for $p = 0.7$). The overlap is calculated as the size of the intersection divided by the size of the smaller set. This metric captures the extent to which the models agree on the tokens that collectively account for the majority of the probability mass.

**Example Mask Construction.**   Consider an input sequence: $[x_1, x_2, x_3, x_4, x_5, x_6, \text{pad}, ...]$, where $(x_2, x_3)$ and $(x_5, x_6)$ are merged. The resulting masks are:

| Position | $x_1$ | $(x_2, x_3)$ | $x_4$ | $(x_5, x_6)$ | pad | ... |
|---|---|---|---|---|---|---|
| Attention Mask | 1 | 1 | 1 | 1 | 0 | ... |
| Loss Mask | 0 | 1 | 1 | 1 | 0 | ... |

---

**Algorithm 1** Masked Cross-Entropy Loss Computation

---

**Require:** Compressed probabilities $\mathrm{CP} \in \mathbb{R}^{B \times S \times V}$, original probabilities $\mathrm{TP} \in \mathbb{R}^{B \times S \times V}$, loss mask $M_{\mathrm{loss}} \in \{0, 1\}^{B \times S}$, attention mask $M_{\mathrm{attn}} \in \{0, 1\}^{B \times S}$

**Ensure:** Average masked cross-entropy loss $\mathcal{L}$

1: For each sample $i$ in batch:
2:     Compute number of valid positions in attention mask $n_{\mathrm{attn}} = \sum_j M_{\mathrm{attn}}[i, j]$
3:     Compute number of valid positions in loss mask $n_{\mathrm{loss}} = \sum_j M_{\mathrm{loss}}[i, j]$
4:     If $n_{\mathrm{attn}} > n_{\mathrm{loss}}$, set first $n_{\mathrm{attn}} - n_{\mathrm{loss}}$ positions of $M_{\mathrm{attn}}[i]$ to 0 to create adjusted mask $M_{\mathrm{adj}}[i]$
5: For each sample $i$, select original probabilities at positions where $M_{\mathrm{loss}}[i] = 1$ and compressed probabilities at positions where $M_{\mathrm{adj}}[i] = 1$
6: Pad all selected probability sequences to fixed length $L$
7: Stack aligned teacher and student probabilities: $T_{\mathrm{aligned}}, S_{\mathrm{aligned}} \in \mathbb{R}^{B \times L \times V}$
8: Compute masked cross-entropy:

$$\mathcal{L} = -\frac{1}{\sum M_{\mathrm{loss}}} \sum_{b,l} \sum_v T_{\mathrm{aligned}}[b, l, v] \cdot \log(S_{\mathrm{aligned}}[b, l, v] + \epsilon)$$

where sum over $(b, l)$ only includes positions with $M_{\mathrm{loss}}[b, l] = 1$.
9: **return** $\mathcal{L}$

---

**Algorithm 2** Masked Top-1 Accuracy Computation

---

**Require:** Teacher probabilities $T \in \mathbb{R}^{B \times S \times V}$, student probabilities $S \in \mathbb{R}^{B \times S \times V}$, teacher mask $M_T \in \{0, 1\}^{B \times S}$, student mask $M_S \in \{0, 1\}^{B \times S}$

**Ensure:** Masked top-1 accuracy $\mathrm{Acc}_{\mathrm{top1}}$

1: For each batch $b$, pad teacher and student sequences to fixed length $L$
2: For each sample $b$:
3:     Select rows in $T[b]$ where $M_T[b] = 1$, pad to $L$; repeat for $S[b]$ and $M_S[b]$
4:     Create mask vectors $F_T[b], F_S[b]$ where 1 marks valid rows
5: Stack all to get $T_{\mathrm{aligned}}, S_{\mathrm{aligned}}, F_T, F_S$
6: Compute top-1 predictions for both: $\mathrm{top1}_T = \mathrm{argmax}_v(T_{\mathrm{aligned}})$, $\mathrm{top1}_S = \mathrm{argmax}_v(S_{\mathrm{aligned}})$
7: Ensure $F_T = F_S$; if not, raise an error
8: Compute combined mask $F = F_T \wedge F_S$
9: Compute match matrix $M = (\mathrm{top1}_T == \mathrm{top1}_S)$
10: $\mathrm{numer} = \sum_{b,l} M[b, l] \cdot F[b, l]$
11: $\mathrm{denom} = \sum_{b,l} F[b, l]$
12: $\mathrm{Acc}_{\mathrm{top1}} = \begin{cases} 0, & \mathrm{denom} = 0 \\ \mathrm{numer/denom}, & \mathrm{otherwise} \end{cases}$
13: **return** $\mathrm{Acc}_{\mathrm{top1}}$

---

**Algorithm 3** Masked Top-$k$ Overlap Computation

---

**Require:** Teacher probabilities $T \in \mathbb{R}^{B \times S \times V}$, student probabilities $S \in \mathbb{R}^{B \times S \times V}$, teacher mask $M_T \in \{0, 1\}^{B \times S}$, student mask $M_S \in \{0, 1\}^{B \times S}$, overlap $k$

**Ensure:** Masked top-$k$ overlap $\mathrm{Overlap}_k$

1: For each sample, select and pad teacher/student logits as in Algorithm 2; align and flatten valid rows
2: For each valid position $m$:
3:     Compute top-$k$ indices for teacher: $I_m^T = \mathrm{TopK}(T_m, k)$
4:     Compute top-$k$ indices for student: $I_m^S = \mathrm{TopK}(S_m, k)$
5:     Count overlap: $c_m = |I_m^T \cap I_m^S|$
6:     Compute per-position ratio: $r_m = c_m/k$
7: $\mathrm{Overlap}_k = \mathrm{mean}_m(r_m)$
8: **return** $\mathrm{Overlap}_k$

---

---

**Algorithm 4** Masked Top-$p$ Overlap Computation

---

**Require:** Teacher probabilities $T \in \mathbb{R}^{N \times V}$, student probabilities $S \in \mathbb{R}^{N \times V}$, mask $M \in \{0, 1\}^N$, threshold $p \in (0, 1)$

**Ensure:** Top-$p$ overlap metric $\text{Overlap}_p$

1: For all positions $i$ where $M[i] = 1$ (i.e., valid tokens):
2:   Sort $T[i]$ in descending order, obtain sorted indices $I_i^T$, values $V_i^T$
3:   Compute cumulative sum $C_i^T$ of $V_i^T$
4:   Find minimal $k_T$ such that $C_i^T[k_T] \geq p$
5:   Define $P_i^T = \{I_i^T[0], ..., I_i^T[k_T]\}$
6:   Repeat above steps for $S[i]$, obtain $P_i^S$
7:   Compute intersection size: $n_{\cap,i} = |P_i^T \cap P_i^S|$
8:   Let $n_{\min,i} = \min(|P_i^T|, |P_i^S|)$
9:   Set overlap ratio $r_i = n_{\cap,i}/n_{\min,i}$
10: $\text{Overlap}_p = \text{mean}_i(r_i)$ over all valid positions $i$
11: **return** $\text{Overlap}_p$

---

**Algorithm 5** Masked Mean Reciprocal Rank (MRR) Computation

---

**Require:** Teacher probabilities $T \in \mathbb{R}^{N \times V}$, student probabilities $S \in \mathbb{R}^{N \times V}$, mask $M \in \{0, 1\}^N$

**Ensure:** Masked MRR MRR

1: For all positions $i$ where $M[i] = 1$:
2:   Find the correct token index: $y_i^* = \text{argmax}_v T[i, v]$
3:   Sort $S[i]$ in descending order, get ranked indices $R_i$
4:   Find rank $r_i$ of $y_i^*$ in $R_i$ (i.e., $R_i[r_i] = y_i^*$)
5:   Compute reciprocal rank: $rr_i = 1/(r_i + 1)$
6: $\text{MRR} = \text{mean}_i(rr_i)$ over all valid positions $i$
7: **return** MRR

---

# D   MODEL INFERENCE WITH UMIM ON-THE-FLY

We provide two algorithms for integrating UMIM into the inference process of pre-trained language models Section 3.4. Rather than being mutually exclusive or strictly alternative, these algorithms offer flexible integration strategies that can be selected based on deployment requirements or resource constraints. Importantly, both algorithms are compatible with a wide range of autoregressive decoding strategies, such as greedy decoding, top-$k$ sampling, and top-$p$ (nucleus) sampling. Our approach does not impose any restrictions on the choice of decoding algorithm; UMIM operates independently of the underlying generation method and can be seamlessly incorporated into any autoregressive inference pipeline. This design ensures broad applicability, allowing practitioners to benefit from token merging and context compression regardless of their preferred decoding configuration.

**Prompt-Only Merging**   In this approach, UMIM is applied a single time to the input prompt prior to the start of generation. Specifically, we scan the entire tokenized prompt for all eligible $n$-grams as defined in the merge set $\mathcal{R}$, and each matching span is replaced with its corresponding surrogate embedding, generated via our multi-head pooling mechanism. This transformation produces a compressed sequence of embeddings that serves as the model's input. The compressed prompt is then fed into the language model, after which generation proceeds in the standard autoregressive fashion without any further merging or intervention by UMIM. Because token merging is performed only once on the initial prompt, this strategy leads to an immediate reduction in both the effective prompt length and the size of the initial key-value (KV) cache required for generation. As a result, prompt-only merging yields significant efficiency gains in terms of memory and computation, all while preserving full compatibility with any standard autoregressive decoding algorithm.

Pseudocode 3: Prompt-Only Merging

```
# Prompt-Only Merging
# Inputs: x (input text), merge rules R, trained NAMEM, pretrained LM
```

```
z = Tokenizer(x)
z_merged = z.copy()

# For each n-gram length, merge eligible spans
for n in [4, 3, 2]:
    while True:
        found = False
        for i in range(len(z_merged) - n + 1):
            span = tuple(z_merged[i:i+n])
            if span in R[n]:
                z_merged = z_merged[:i] + [span] + z_merged[i+n:]
                found = True
                break
        if not found:
            break

# Convert tokens/tuples to embeddings
E = []
for element in z_merged:
    if isinstance(element, tuple):
        emb = multi_head_pooling(element)  # Surrogate embedding
    else:
        emb = model_embedding(element)
    E.append(emb)

E = stack(E)
y = LM.generate(inputs_embeds=E, ...)
return y
```

**Dynamic Online Merging During Generation**   In the second approach, UMIM is applied dynamically throughout autoregressive decoding. After each new token is generated, we examine whether the most recent n-gram (ending with the new token) matches any entry in $\mathcal{R}$. If a match is found, we roll back the last $n$ steps: removing their KV cache entries and replacing them with a single surrogate embedding produced by UMIM. This embedding is re-encoded and reinserted into the cache, allowing the model to continue generation with a compressed sequence. This on-the-fly merging strategy incrementally reduces context length, minimizing memory and computation cost as generation progresses.

Pseudocode 4: Dynamic Online Merging

```
# Dynamic Online Merging
# Inputs: x (input text), merge rules R, trained NAMEM, pretrained LM

z = Tokenizer(x)
z_merged = z.copy()

# Initial prompt merging
for n in [4, 3, 2]:
    while True:
        found = False
        for i in range(len(z_merged) - n + 1):
            span = tuple(z_merged[i:i+n])
            if span in R[n]:
                z_merged = z_merged[:i] + [span] + z_merged[i+n:]
                found = True
                break
        if not found:
            break

# Initialize embeddings and dynamic KV cache
E = []
for element in z_merged:
    if isinstance(element, tuple):
```

```
        emb = multi_head_pooling(element)
    else:
        emb = model_embedding(element)
    E.append(emb)
E = stack(E)
C = init_kv_cache(E)
logits = compute_logits(E, C)

# Autoregressive generation with dynamic merging
while not finished:
    y = sample_from_logits(logits)
    z.append(y)
    z_merged.append(y)

    # Check for eligible merges with new token
    for n in [4, 3, 2]:
        if len(z_merged) >= n:
            span = tuple(z_merged[-n:])
            if span in R[n]:
                z_merged = z_merged[:-n] + [span]
                C = remove_last_n_from_cache(C, n)
                emb = multi_head_pooling(span)
                C = insert_embedding_in_cache(C, emb)
                break

    logits = compute_logits(z_merged, C)

output = decode(z_merged)
return output
```

# E    IMPLEMENTATION DETAILS

## E.1    TRAINING DATA SETS

To train our merge module across various base models—including Llama 3.1 8B, Llama 3.2 1.5B, and GPT2-XL 1.5B—we utilize the WikiText-103 corpus. We preprocess all texts and concatenate them into a fixed-length dataset, where each sequence has a length of 512 tokens. For the DeepScaleR-1.5B-Preview model, we use a custom-generated dataset. Specifically, we leverage questions from two prominent reasoning benchmarks: KbsdJames/Omni-MATH Gao et al. (2024) and RUC-AIBOX/STILL-3-Preview-RL-Data Jiang et al. (2024). For each question, we construct a prompt consisting of the question text and then perform three independent sampling runs, generating additional tokens such that the combined length of the prompt, question, and generated output reaches 1024 tokens per sample. All samples are concatenated to form a fixed-length dataset, where each sequence is exactly 1024 tokens.

## E.2    PARAMETER SETTINGS AND TRAINING IMPLEMENTATION CHOICES

**General Implementation Details**    All experiments were conducted on 8 NVIDIA GPUs (a mix of RTX A6000, RTX 6000 Ada, and L40S, each with 48GB memory). For distributed training, either PyTorch DistributedDataParallel (DDP) or DeepSpeed ZeRO stage 2 in bf16 precision was used, depending on the model. The optimizer was AdamW (learning rate $8 \times 10^{-4}$, weight decay $1 \times 10^{-3}$), with learning rate warmup and decay. Each model was trained for 15 epochs. The datasets were preprocessed, saved as `.pt` or `.pkl` files, and loaded via a custom `TextDataset` with `DistributedSampler`. The teacher model for distillation was frozen, and only the merge module parameters in the student model were updated. Checkpoints were saved after each epoch, with the best model selected by validation loss and early stopping (patience = 3). All metrics—including cross-entropy loss, top-1/top-3/top-10 accuracy, top-$p$ ($p = 0.7$), and MRR—were logged with Weights & Biases (wandb).

**Model-Specific Settings**

- **Llama 3.1-8B**: DeepSpeed ZeRO stage 2, batch size 3 per GPU, sequence length 512, merge module $d_{\mathrm{model}} = 4096$.

- **Llama 3.2-1.5B**: DDP, batch size 4 per GPU, sequence length 512, merge module $d_{\mathrm{model}} = 2048$.

- **GPT2-XL**: DDP, batch size 4 per GPU, sequence length 512, merge module $d_{\mathrm{model}} = 1600$.

- **DeepScaleR-1.5B-Preview**: DeepSpeed ZeRO stage 2, batch size 3 per GPU, sequence length 1024, merge module $d_{\mathrm{model}} = 1536$.

## F  ADDITIONAL EXPERIMENTS AND EVALUATION

### F.1  TEXT GENERATION

#### F.1.1  TEXT GENERATION SETUP

We describe our experimental setup for text generation with Llama 3 8B:

For qualitative results, we randomly select five prompts from the WikiText-103 test split. For each prompt, we generate either 20 or 30 tokens, depending on the length required to illustrate a complete sequence, using greedy decoding. Additionally, we select one prompt from the WikiText-103 test split and perform top-$p$ sampling with $p = 0.7$ and temperature $0.8$. For this prompt, we sample three independent generations.

For language modeling performance, we use WikiText-103, BookCorpus, and OpenWebText. We directly compute the perplexity for both the base model and our merge model on these datasets. The PPL calculation process for the merge module is described in(See algorithm 5).

For quantitative results, we use the WikiText-103 test split and report metrics including Rep-2, Dist-2, Rep-3, Dist-3, Rep-4, Dist-4 Li et al. (2016), ROUGE-2, ROUGE-L, ROUGE-Lsum, BERTScore, chrF Popović (2015), and Mauve Pillutla et al. (2021). For each sample, the first 100 tokens are used as the prompt, and the next 100 tokens (tokens 101–200) as the reference. The model is conditioned on the prompt to generate 100 tokens, which are then compared against the reference sequence. We evaluate three decoding strategies: greedy decoding, top-$p$ sampling (with $p = 0.7$, temperature $= 0.8$), and random sampling. For both top-$p$ and random sampling, we perform five independent generation runs for each input and report the average results.

Pseudocode 5: Two-Stage PPL Calculation for Merge Model

```
# Two-Stage PPL Calculation for Dynamic Merge Model

# Inputs:
#   x: token id sequence, length N+1
#   model: dynamic merge model (with pooling module etc)
#   merge_rules: {2: bigrams, 3: trigrams, 4: fourgrams}

# Step 0: Merge sequence and record skipped relations
merged_seq, merge_records = merge_with_context(x, merge_rules)
# merged_seq: [token_id or tuple], e.g. [x1, (x2,x3), x4, (x5,x6), x7]
# merge_records: dict, e.g. {2: {'context': [...], 'current_token': x2, '
    target_token': x3}, ...}

# Step 1: Merged-sequence loss
L1 = 0
inputs = merged_seq[:-1]
targets = []
for i in range(len(inputs)):
    next_elem = merged_seq[i+1]
    if isinstance(next_elem, tuple):
        targets.append(next_elem[0])  # use first token of merged tuple
    else:
        targets.append(next_elem)
input_embs = []
for elem in inputs:
```

```
1350        if isinstance(elem, tuple):
1351            emb = pool_embedding(elem)    # pool embedding for merged n-gram
1352        else:
1353            emb = embedding(elem)
1354        input_embs.append(emb)
1355 logits = model.forward(stack(input_embs))
1356 for t, target in enumerate(targets):
1357     L1 += -log_prob(logits[t], target)
1358 # Step 2: Skipped-token loss
1359 L2 = 0
1360 for pos, rec in merge_records.items():
1361     ctx = rec['context']
1362     current = rec['current_token']
1363     target = rec['target_token']
1364     ctx_embs = []
1365     for c in ctx:
1366         if isinstance(c, tuple):
1367             emb = pool_embedding(c)
1368         else:
1369             emb = embedding(c)
1370         ctx_embs.append(emb)
1371     ctx_embs.append(embedding(current))
1372     logits = model.forward(stack(ctx_embs))
1373     L2 += -log_prob(logits[-1], target)
1374
     # Step 3: Total loss and Perplexity
     L_total = L1 + L2
     PPL = exp(L_total / (len(x)-1))
```

### F.1.2 QUANTITATIVE RESULTS

We present comprehensive qualitative generation results as shown in Tables 8 and 9. Across all decoding strategies, we observe that the merge module can effectively condense the semantic content of multiple tokens into a single surrogate token, with minimal impact on the fluency or coherence of the generated text.

Importantly, the merged tokens are not simply unimportant or redundant. During autoregressive generation, merged tokens retain substantial influence, as evidenced by their significant attention weights in the model's output. This indicates that the model continues to rely on merged tokens for contextual reasoning and content prediction, rather than disregarding them. Visualization of attention distributions further supports that merged tokens play an active and meaningful role in the generative process, as indicated in Figure 3.

In addition, embedding similarity analysis shows that the representation of a merged token exhibits near-zero cosine similarity with the embeddings of its original constituent tokens. This demonstrates that the merge module learns to encode a distinct, context-aware representation for each merged span.

### F.1.3 QUANTITATIVE RESULTS

Quantitative metrics—including n-gram repetition, diversity (Rep-$n$, Dist-$n$), and multiple text similarity scores (ROUGE, BERTScore, chrF, Mauve)—confirm that the merge model maintains generation quality comparable to the base model across all decoding settings. Overall, our results demonstrate that the merge module enables effective semantic compression, while preserving the expressiveness, interpretability, and attention dynamics of the underlying language model.

Table 8: Qualitative generation results with greedy decoding. Highlighted texts are merged with UMIM, with [] for merging boundaries.

| Prompt | Generation | |
| --- | --- | --- |
| | **Base Model** | **Merge Module** |
| Traditional Chinese literary criticism emphasized **[the life of the]** author when interpreting a work | The author's life was seen as the key to understanding the work. | The author's life **[was seen as]** the key **[to understanding the work]**. |
| In **[2025]**, **[he moved to the]** institute **[in an attempt]** to study. | He was a very good student, and he was very good at his job. | He **[was a student of]** **[the institute and]** **[was a member of]** the institute's football team. |
| In the spring, **[his youngest son]**, Ian, was born. | He was a beautiful baby boy, and he was the light of his father's life. | He **[was a beautiful baby]**, **[with a head of thick]**, dark **[hair and]** a round, chubby face. |
| The first public library in Bintulu **[was built in]** 1971. | It was located at the Bintulu District Office. The library was later moved to the Bintulu Town Council building in 1973. | It **[was located at]** the Bintulu District Office. The library was **[later moved to the]** Bintulu Town Council building **[in 1973]**. |
| Everyone considered the reviewers **[an improvement on the]** conference. | The reviewers were very helpful and the reviews were very thorough. The reviewers were very helpful and the reviews were very thorough. | The reviewers were very professional **[and the conference]** was very well organized. The conference **[was a great success]** and we are looking forward to next year's conference. |

Table 9: Qualitative generation results with Top-p decoding.Highlighted texts are merged with UMIM, with [] for merging boundaries.

| **Prompt:** Tell me something about Lakers. | | | |
| --- | --- | --- | --- |
| | Sample 1 | Sample2 | Sample3 |
| Base | Why did they change their name from Minneapolis Lakers to Los Angeles Lakers? What are their nicknames? What are their logos? Who are their rivals? Who are their greatest players? What are their colors? What is their motto? | What do you think of Kobe? Lakers are the best team in the league, I believe. I think they will win the championship again. Kobe is the best player in the league, and I think he will win the MVP this year. | I've heard that they are the best team in the NBA, but I'm not sure about that. Who are the best players on the team? What are their chances to win the NBA championship this season? I want to see them in action. |
| Merge | Tell me something about Lakers. I never **[been to a]** game. I want **[to know everything]** about **[the team and the]** players **[and the history of]** the team. I'm like **[a kid in]** **[a candy store]**. **[I just want to]** know everything. | What **[do you think of]** their chances **[to make the playoffs]** this year? What **[do you think of]** **[the Clippers]**? Do you think they'll be **[able to win the]** championship this year? What **[do you think of]** the Lakers' chances? | What is **[the most exciting thing]** about Lakers? Why do you love them? What makes **[them special]**?The Lakers **[are a very]** special team. They have had **[some of the greatest]** players **[to ever play the]** **[game of basketball]**. |

Table 10: Quantitative results of N-gram repetition and diversity metrics

| Method | Rep-2 | Dist-2 | Rep-3 | Dist-3 | Rep-4 | Dist-4 |
|---|---|---|---|---|---|---|
| Top-$p$ (Base model) | 11.76 | 39.58 | 10.93 | 70.14 | 6.78 | 85.59 |
| Top-$p$ (Merge Model) | 11.28 | **43.49** | 9.08 | **75.59** | 4.55 | **89.92** |
| Greedy (Base model) | 14.03 | 40.99 | 15.32 | 60.38 | 13.97 | 69.75 |
| Greedy (Merge Model) | 13.34 | 39.04 | 14.44 | 58.96 | 12.74 | 69.30 |
| Sampling (Base model) | 11.06 | 50.72 | 7.13 | 83.10 | 2.84 | 94.68 |
| Sampling (Merge Model) | 10.77 | 33.96 | **12.02** | 59.34 | 9.76 | 74.25 |

Table 11: Quantitative results of text similarity

| Method | Rouge-2 | Rouge-L | Rouge-Lsum | BERTScore | chrF | Mauve |
|---|---|---|---|---|---|---|
| Top-$p$ (Base model) | 5.49 | 17.85 | 19.66 | 83.75 | 28.53 | 34.87 |
| Top-$p$ (Merge Model) | 4.87 | 17.17 | 19.14 | 83.57 | 28.39 | 33.90 |
| Greedy (Base model) | 5.82 | 18.89 | 20.03 | 83.35 | 26.99 | 64.28 |
| Greedy (Merge Model) | 5.38 | 18.44 | 19.47 | 82.98 | 26.08 | **67.21** |
| Sampling (Base mode) | 4.12 | 16.23 | 18.24 | 83.25 | 28.28 | 36.81 |
| Sampling (Merge Model) | **5.45** | **18.09** | **19.58** | **83.44** | 27.67 | 27.41 |

## F.2 DOWNSTREAM TASKS

We strictly follow the framework of LLM Evaluation test QA. We combine prompts, questions, and answers into a sequence. Then we calculate the loss of each overall context. We select the option with the lowest loss as the correct answer and compare it with the correct answer.

For the CNN/DailyMail task, we conducted experiments using the LLaMA-8B model. Consistent with previous setups, token merging was guided by merge rules identified from the WikiText training split. All outputs were generated using greedy decoding with a fixed length of 80 tokens. To ensure comparability with baselines, token merging was applied exclusively to the input portion of each example.For the Select Context baseline, token-level reduction of the prompt was performed. Regarding LLMLingua1, we employed its default token reduction method, while for LLMLingua2, we utilized its small model variant for prompt reduction. For the Sink Attention baseline, we configured 4 sink tokens and a sliding window equal to 16% of the total length (prompt length plus generated tokens). Due to the inherent difficulty in precisely controlling the token reduction with LLMLingua and Select Context, their reported token reductions are approximate. In contrast, our approach ensures precise control over the exact number of merged tokens.The experimental results clearly demonstrate that our token merging method consistently outperforms all baselines, even in a domain entirely unseen during training. This confirms not only the strong merging capability of our module but also highlights its generalization and robustness across downstream tasks beyond the original training domain. We conducted evaluations on the AIME2024 and AMC mathematics benchmarks using the DeepScaleR model. Given that reasoning tasks typically require generating a large number of tokens, computational resource limitations prevented extensive training involving such lengthy outputs. Therefore, token merging was applied exclusively to the input prompts. We set the maximum generation length to 15,000 tokens and assessed performance using the pass@k(k=16) accuracy metric.Experimental results indicate that our merge module consistently achieves strong performance on these challenging reasoning tasks. Notably, our method outperforms both Select Context and LLMLingua baselines, further underscoring its effectiveness and generalizability in reasoning-intensive applications.

## F.3 EFFICIENCY ANALYSIS

While GPUs provide massive parallel compute capacity, the primary bottleneck in autoregressive decoding is not arithmetic throughput but I/O. Due to the sequential nature of Transformers, each decoding step only generates a single new token, so the raw computation per step is relatively small. However, every new token must attend to the entire accumulated KV cache, which requires repeatedly

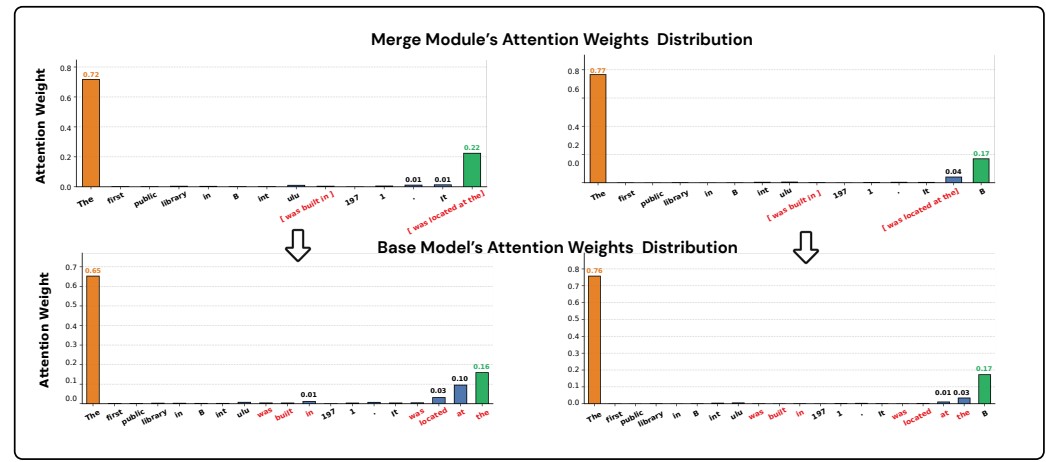

Figure 3: Attention distribution after applying the merge module at inference compared with the original token-based inference.

reading large memory blocks. This KV cache access dominates runtime and thus becomes the key limiter of both efficiency and GPU memory usage. Therefore, reducing the effective cache size—through prompt compression and on-the-fly merging during decoding—is the most direct way to improve efficiency.

**Baseline.** Huggingface's API achieves high throughput because the decoding pipeline—including embedding lookup, attention computation, KV cache updates, and token sampling—is fully fused into highly optimized C++/CUDA kernels. All computations run on GPU end-to-end with minimal kernel launches and essentially no CPU–GPU synchronization overhead.

**Our prototype.** In contrast, our current implementation is controlled by Python logic. Token merging, embedding pooling, and KV cache management are all handled at the interpreter level, which introduces overhead from scheduling and frequent CPU–GPU synchronizations. Each merge or cache update triggers separate, small kernel launches. As a result, our measured latency improvement is only marginal and does not yet reflect the theoretical acceleration of our approach.

**Conclusion.** This is purely an *implementation limitation*, not a flaw of the method. Once the merging operations are fused into optimized GPU kernels and the Python overhead is removed, we expect the observed latency speedup to closely match the efficiency gains predicted by our FLOPs and memory analysis Table 12.

### F.3.1 FLOPs and Memory Analysis

Let $N$ be the prompt length, $n$ the number of generated tokens, $D$ the hidden dimension, $L$ the number of transformer layers, and $C$ the KV cache entry size.

**Prompt merging.** Merging a fraction of the prompt (e.g., 50%) directly reduces the input length from $N$ to $N/2$, which immediately halves the memory footprint of the KV cache and lowers the computational cost of attention for the entire generation process.

**Generation merging.** During autoregressive decoding, merging spans provides *cumulative savings*. For example, with bigram merging (every two generated tokens merged into one surrogate), the cumulative attention cost reduces from

$$A = \sum_{t=1}^{n} f(N + t)$$

| Setting | Model | Latency (s) | Throughput (tok/s) |
|---|---|---|---|
| 1000+1000, Merge 50% | Baseline-3.1-8B | 46.10 | 21.69 |
| | Dynamic-8B | 45.34 | 22.05 |
| 2000+2000, Merge 50% | Baseline-3.1-8B | 93.07 | 21.48 |
| | Dynamic-8B | 90.69 | 22.05 |

Table 12: Latency and throughput comparison between the Huggingface baseline (`generate()`) and our dynamic merging prototype. Despite being implemented in Python with significant interpreter overhead, our method already achieves lower latency and higher throughput. This suggests that once merged into fused GPU kernels, the realized speedups will substantially exceed the current measurements.

to

$$B = \sum_{t=1}^{n/2} f(N + t),$$

where $f(\cdot)$ denotes the per-step attention FLOPs. The savings $(A - B)$ grow quadratically with $n$, while KV cache usage is also reduced by roughly 50%.

**Interpretation.** Prompt merging yields *immediate one-time savings*, while generation merging yields *cumulative quadratic savings* as decoding length increases. Both effects directly reduce I/O costs by shrinking the KV cache.

**Summary.** In summary, the merge module replaces $n$-gram spans with surrogate embeddings, thereby shortening the effective sequence length without altering the underlying model semantics. This yields immediate benefits during prompt processing, where the reduced input directly lowers KV cache size and attention FLOPs from the very beginning of generation. During autoregressive decoding, the effect compounds: each merge incrementally shrinks the cache and reduces computation, leading to quadratic savings as the sequence grows longer. Although our current latency measurements are limited by Python-level implementation overhead, these results do not reflect a weakness of the method itself. Once the pipeline is fused into optimized GPU kernels, the realized acceleration is expected to align closely with the theoretical gains predicted by our FLOPs and memory analysis.

## G ABLATION STUDY

### G.1 DIFFERENT THRESHOLDS OF N-GRAMS

In summary, our results clearly show that as the proportion of merged tokens increases, the training metrics consistently decrease. This relationship is direct and inevitable: a lower merge threshold $\tau$ leads to more frequent merging, which in turn results in a larger portion of the sequence being combined into n-grams and a corresponding decline in model performance. This trend is observed across both Llama-3.2-1.5B and GPT2-XL (See Tables 13 and 14). These findings highlight an inherent trade-off: while merging more tokens can improve efficiency by reducing sequence length, it unavoidably comes at the cost of lower training quality. Therefore, selecting an appropriate merge threshold is essential to balance efficiency and performance in practice.

Table 13: Different Threshold of N-grams -Llama 3.2 1.5B

| $\tau$ | Top1 | Top3 | Top10 | Top p | MRR |
|---|---|---|---|---|---|
| 500 | 0.8437 | 0.8356 | 0.8433 | 0.9504 | 0.9078 |
| 50 | 0.7883 | 0.7840 | 0.7943 | 0.9320 | 0.8691 |
| 5 | 0.7410 | 0.7414 | 0.7528 | 0.9155 | 0.8297 |

Table 14: Different Threshold of N-grams GPT2 XL

| $\tau$ | Top1 | Top3 | Top10 | Top p | MRR |
|---|---|---|---|---|---|
| 500 | 0.8414 | 0.8456 | 0.8579 | 0.9548 | 0.9048 |
| 50 | 0.7588 | 0.7707 | 0.7893 | 0.9297 | 0.8447 |
| 5 | 0.6955 | 0.7128 | 0.7359 | 0.9065 | 0.7928 |

## G.2 DIFFERENT MERGE MODULE SIZES

To assess the impact of model capacity, we conduct an ablation study on the size of the merge module. Specifically, we retrain merge modules of different depths on WikiText-103, fixing $\tau$ at 5. As shown in Table 17, increasing the number of attention layers from one to three results in only marginal performance improvements across all evaluation metrics. These results indicate that our merge module design is highly parameter-efficient and already achieves strong performance in this scenario.

Table 15: Different Merge Module Size Based on Llama-3.2-1.5B

| Merge Module Size | $\tau$ | Top-1 | Top-3 | Top-10 | Top-$p$ | MRR |
|---|---|---|---|---|---|---|
| One Attention Layer | 5 | 74.10 | 74.14 | 75.28 | 91.55 | 83.30 |
| Three Attention Layers | 5 | 74.27 | 74.15 | 75.34 | 91.58 | 83.41 |

## G.3 DIFFERENT DECODING ALGORITHMS GENERATION RESULTS

We employ various decoding strategies, including greedy decoding, top-p sampling, and random sampling, to generate context. The corresponding results are presented in Section F. Overall, our approach does not lead to substantial losses in generation quality.

## G.4 DIFFERENT MERGE RULES

We also propose an alternative approach for constructing merge rules by leveraging a language model (Llama-3.2-1.5B) to compute the generation probability of each token within input text chunks. Specifically, we automatically extract all consecutive spans of tokens whose probabilities exceed a predefined threshold (set to 0.5) and are at least two tokens in length. These high-confidence token sequences are collected as merge rules, representing common and reliable language patterns as identified by the model. To ensure compatibility with our decoding algorithm, we discard any candidate chunk longer than four tokens (i.e., restrict merges to at most four-grams). As a result, the total number of TR is relatively modest. According to training alignment metrics, the merge rules constructed using the method described in CD-LM also yield strong training results. While our method is practical and effective, it is not the only possible strategy—other approaches for constructing merge rules may also be considered.

Table 16: CD LM Merge Rules

| Model | TR (%) | Top-1 | Top-3 | Top-10 | Top-$p$ | MRR |
|---|---|---|---|---|---|---|
| Llama-3.2-1.5B | 10.3% | 91.07 | 91.04 | 91.52 | 97.17 | 94.96 |

## G.5 DIFFERENT SIZES OF TRAINING DATA

We extend the training corpus by combining WikiText and BookCorpus, resulting in a dataset that is approximately three times larger than the original. We then train the merge module for Gemma3 4B on this enlarged dataset. Intuitively, with the same training corpus, a larger merge module (i.e., more parameters) should yield better performance. However, we observe that the 4B model does not outperform the 1.5B model across evaluation metrics. This suggests that the effectiveness of training is also influenced by the size of the dataset.

Table 17: Predictive distribution alignment metrics, with token reduction (TR) rate.

| Model | TR (%) | Top-1 | Top-3 | Top-10 | Top-$p$ | MRR |
|---|---|---|---|---|---|---|
| Llama-3.1-8B | 37.8% | 79.22 | 77.56 | 77.58 | 92.91 | 87.05 |
| Llama-3.2-1.5B | 37.8% | 74.10 | 74.14 | 75.28 | 91.55 | 83.30 |
| GPT2-XL | 35.74% | 69.55 | 71.28 | 73.59 | 90.65 | 79.28 |
| Gemma3 4B | 39.7% | 70.41 | 70.72 | 72.35 | 89.39 | 85.59 |
| DeepScaleR-1.5B-Preview | 54% | 77.00 | 69.30 | 67.99 | 90.98 | 85.59 |

## ADDITIONAL REFERENCES FOR APPENDIX

Bofei Gao, Feifan Song, Zhe Yang, Zefan Cai, Yibo Miao, Qingxiu Dong, Lei Li, Chenghao Ma, Liang Chen, Runxin Xu, Zhengyang Tang, Benyou Wang, Daoguang Zan, Shanghaoran Quan, Ge Zhang, Lei Sha, Yichang Zhang, Xuancheng Ren, Tianyu Liu, and Baobao Chang. Omni-math: A universal olympiad level mathematic benchmark for large language models, 2024. URL https://arxiv.org/abs/2410.07985.

Jinhao Jiang, Zhipeng Chen, Yingqian Min, Jie Chen, Xiaoxue Cheng, Jiapeng Wang, Yiru Tang, Haoxiang Sun, Jia Deng, Wayne Xin Zhao, Zheng Liu, Dong Yan, Jian Xie, Zhongyuan Wang, and Ji-Rong Wen. Enhancing llm reasoning with reward-guided tree search, 2024. URL https://arxiv.org/abs/2411.11694.

Jiwei Li, Michel Galley, Chris Brockett, Jianfeng Gao, and Bill Dolan. A diversity-promoting objective function for neural conversation models, 2016. URL https://arxiv.org/abs/1510.03055.

Krishna Pillutla, Swabha Swayamdipta, Rowan Zellers, John Thickstun, Sean Welleck, Yejin Choi, and Zaid Harchaoui. Mauve: Measuring the gap between neural text and human text using divergence frontiers, 2021. URL https://arxiv.org/abs/2102.01454.

Maja Popović. chrF: character n-gram F-score for automatic MT evaluation. In Ondřej Bojar, Rajan Chatterjee, Christian Federmann, Barry Haddow, Chris Hokamp, Matthias Huck, Varvara Logacheva, and Pavel Pecina (eds.), *Proceedings of the Tenth Workshop on Statistical Machine Translation*, pp. 392–395, Lisbon, Portugal, September 2015. Association for Computational Linguistics. doi: 10.18653/v1/W15-3049. URL https://aclanthology.org/W15-3049/.

