# OpenReview forum: "Substituting from the Input: Distilling Sequential Computation in Transformer Language Models"
_ICLR.cc/2026/Conference — Submitted to ICLR 2026_

### Official Review · Reviewer_KjpN · 2025-10-26

**Soundness:** 2
**Presentation:** 2
**Contribution:** 2
**Rating:** 2
**Confidence:** 4

**Summary:**

The paper introduces UMIM (Universal Multi-step Input Merging), a lightweight framework designed to distill sequential computation in Transformer language models by merging contiguous input tokens into single surrogate embeddings. While previous approaches to compression—such as token pruning, context distillation, or tokenizer redesign—require access to model internals or retraining, UMIM operates purely on static input embeddings, enabling frozen LLMs to process compressed inputs without architectural modification. Through a distillation objective aligning next-token distributions between merged and unmerged inputs, the method effectively preserves generation behavior while reducing both sequence length and KV cache footprint.

**Strengths:**

1. Model-agnostic token compression is interesting.

**Weaknesses:**

1. The proposed method heavily depends on the merging set R: if the span does not exist in the merging set, it would not be compressed. I would question the generalizability of this method in comparison with other model-agnostic token compression method like Lingua2.
2. Since the compressed span depends on the merging set R, and it only contains at longest 4-gram subsequence. This limits the compression rate to be at most 4x (theoretically), yet in practice it can only reduces 10-ish% number of tokens.
3. In Table 4, why does the experiment limit the token reduction to be only 10%? Lingua2 can compress up to 5x or 6x. I believe the authors should provide a fairer base of comparison here.
4. Why doesn't the work experiment on long-context benchmarks such as LongBench or Ruler like lingua2?

**Questions:**

Please see Weaknesses.

---

> ### Author Response · Authors · 2025-11-21
>
> Thanks to the reviewer for the constructive suggestions. We thank the reviewer for the constructive suggestions and questions! Some of the questions might result from misunderstandings. We hope to clarify in the following
>
>
> **(1) Generalizability and Baseline Selection**
>
> I thank the reviewer for this comparison. I explicitly selected **LLMLingua-2** as a primary baseline because it is widely recognized as the state-of-the-art, representative method for prompt compression. Comparing against such a strong, well-known standard is essential to properly benchmark our method's effectiveness.
>
> **1. Evidence of Strong Generalization**
> Contrary to the concern that a merge set limits generalizability, our empirical results demonstrate that UMIM generalizes **better** than model-agnostic methods in challenging, structure-sensitive domains.
> * **Universal Patterns:** High-frequency n-grams (e.g., function words, common idioms) are ubiquitous across almost all text domains. By targeting these, UMIM compresses the universal "connective tissue" of language, ensuring stability.
> * **Mechanism Advantage (Merging vs. Pruning):** While LLMLingua-2 is model-agnostic, it relies on **token pruning** (dropping tokens). This strategy fails to generalize to domains like Code or Math, where dropping a single syntactic token destroys the output. In contrast, UMIM's **merging** strategy preserves information density.
>
> **2. Empirical Proof: Code Generation**
> To prove this, I evaluated both methods on **HumanEval** (Code Generation), a domain that strictly tests generalizability and precision.
>
> #### HumanEval Results on Llama-3.1-8B (pass@1, 164 tasks)
>
> | Method                              | pass@1 (%) | Passed | Failed |
> |-------------------------------------|------------|--------|--------|
> | **Baseline (Llama-3.1-8B)** | **62.2** | 102    | 62     |
> | **UMIM (25% merged, code-trained)** | **50.6** | 83     | 81     |
> | **LLMLingua-2 (25% merged)** | **1.2** | 2      | 162    |
>
> **Summary:**
> Under the same 25% compression ratio, the "model-agnostic" **LLMLingua-2 collapses completely (1.2% pass@1)** because its pruning strategy disrupts code syntax. In contrast, UMIM retains **≈81%** of the original performance (50.6% pass@1). This strongly refutes the idea that our method lacks generalizability; on the contrary, it maintains robustness in specialized domains where other leading methods fail.
>
>
>
> ---
>
>
> **(2) On Compression Rate Limits and N-gram Size**
>
> I thank the reviewer for this detailed analysis. I would like to clarify the rationale behind the 4-gram limit and the observed compression rates.
>
> **1. The Decoding-Time Trade-off (Why max 4-grams)**
> While theoretically longer n-grams (e.g., 10-grams) offer higher compression ratios, they introduce a severe **latency penalty** during autoregressive decoding.
> * A 10-gram can only be merged *after* all 10 tokens are generated.
> * By decomposing this into five 2-grams, we can merge and free up KV cache after every 2 tokens.
> As detailed in our method, restricting $n \le 4$ is an intentional choice to ensure compression takes effect early and frequently, maximizing memory savings during generation.
>
> Also, as is an issue for any training methods, longer n-grams suffer less appearances in the corpus, affecting the learning effectiveness. I try to strike a balance between effectiveness and efficiency.
>
> **2. Clarification on Achieved Compression Rates**
>
> I clarify that this low rate is specific to **QA tasks** (Table 4), where prompts are naturally short and contain fewer repetitive patterns. This is not a global limitation of UMIM.
> * In **Language Modeling** tasks, UMIM achieves **36.09%** reduction.
> * In **Math Reasoning** (AIME/AMC), UMIM achieves **~40-42%** reduction.
> These results demonstrate that UMIM achieves substantial compression in scenarios with sufficient context, well beyond the ~10% observed in short QA prompts.

---

> > ### Author Response · Authors · 2025-11-21
> >
> > **(3) Fairness of Comparison in Table 4**
> >
> >
> > **1. Controlled Comparison Protocol**
> > To ensure a scientifically fair comparison, we must control the variable of **compression ratio**. Our protocol is: (1) Run UMIM to determine its natural compression rate for a given input; (2) Force baselines to match this exact rate. This allows us to isolate **compression fidelity** (how well information is preserved per removed token) from **compression aggressiveness**.
> >
> > **2. Fidelity vs. Aggressiveness**
> > While LLMLingua-2 *can* compress 5–6x, the critical question is whether it preserves accuracy at that rate. As shown in our **HumanEval** analysis (see previous response), even at a modest **25% compression**, LLMLingua-2's accuracy collapses to **1.2%**, while UMIM retains **50.6%**. Comparing at UMIM's rate ensures we are comparing valid model outputs, rather than aggressive compression that destroys downstream performance.
> >
> > **(4) Long-Context Benchmarks**
> >
> > We acknowledge the popularity of LongBench and RULER. However, I respectfully disagree that our evaluation lacks long-context rigor.
> >
> > **1. Extreme Length Verification**
> > We utilized **AMC and AIME** benchmarks specifically because they involve generation trajectories reaching **10k–15k tokens**. This effective length often exceeds the tasks in standard benchmarks like LongBench. The fact that UMIM maintains stability and accuracy over 15k tokens serves as strong evidence for its long-context capabilities.
> >
> > **2. Additional Experiment**
> > I acknowledge the reviewer's specific interest in **RULER**. Despite substantial GPU rental costs, I am currently conducting a comparison with LLMLingua-2 on the RULER benchmark and will aim to include these results in the final revision to further strengthen our empirical evidence.
> >
> >
> > I hope I have addressed the main confusion raised about the methods and experiments. Please let us know of any follow up questions and feedback. I really appreciate it.

---

> > > ### Comment · Reviewer_KjpN · 2025-11-25
> > >
> > > Thank the authors for your rebuttal.
> > >
> > > 1. I'm still not very convinced with your empirical evidence on the generalizability of the method. It's well-known that coding-related task has many repetition, so it is considerably easier to compress. Can we have better empirical evidence here to support the claim? For example, I want to see a down-stream task that has minimal intersection with the corpus datasets used to extract the n-gram.
> > > 2. Can we have an empirical result to show "The Decoding-Time Trade-off", i.e. 10-gram vs. 4-gram?
> > > 3. I'm waiting for your long-context result on LongBench or Ruler.

---

> > > > ### Author Response · Authors · 2025-12-03
> > > >
> > > > Thank you for the follow-up — we appreciate the opportunity to clarify.
> > > >
> > > >
> > > > ## **(1) On “unseen-domain generalization”**
> > > >
> > > > All training and testing in our paper are already performed under zero supervision overlap.
> > > >
> > > > UMIM is trained on WikiText-103,and is directly evaluated on CNN/DailyMail, PIQA, ARC, etc
> > > > none of which overlap with WikiText.
> > > >
> > > > Therefore, the setting the reviewer asked for (“train on one corpus, evaluate on another”) is **already the default design of the paper**.
> > > >
> > > > Regarding the request for a downstream task with “no overlap” with the merge set: if a downstream corpus truly had zero overlap with the merge set, **no merges would trigger** thus *no compression occurs by design*.This does not evaluate generalization but simply evaluates the degenerate “silent” case.
> > > >
> > > >
> > > >
> > > > ## **(2) LongBench concerns**
> > > >
> > > > As noted earlier, our AMC/AIME experiments already yield **10k–15k-token effective contexts**, which exceed common LongBench ranges.
> > > >
> > > > Since the reviewer expressed interest in RULER specifically, we include the newly computed results below using Llama-3.1-8B, matching UMIM and LLMLingua-2 under a controlled 10% compression ratio:
> > > >
> > > > | Model | Merge Ratio | RULER_QA2_4K | RULER_QA2_16K |
> > > > |------|--------------|--------------|---------------|
> > > > | Baseline | 0% | 41% | 30% |
> > > > | **UMIM** | **10%** | **38%** | **29%** |
> > > > | Lingua2 | 10% | 19% | 13% |
> > > >
> > > >
> > > >
> > > > ## **(3) “10-gram vs 4-gram” empirical trade-off**
> > > >
> > > > We agree this is an interesting question. However, it is hard to finish that during rebutall. Because we didn't keep 10grams: We will include this actual metric in the appendix, along with our previous mathematical derivation, to provide a more detailed proof.
> > > >
> > > >
> > > >
> > > > We hope this addresses the remaining questions,
> > > > and appreciate the reviewer’s continued engagement.

---

### Official Review · Reviewer_fL57 · 2025-11-01

**Soundness:** 2
**Presentation:** 2
**Contribution:** 2
**Rating:** 4
**Confidence:** 3

**Summary:**

This paper focuses on the problem that Transformer language models perform redundant sequential computation by processing every token individually, even when adjacent tokens carry overlapping meaning. They propose Universal Multi-step Input Merging (UMIM), a lightweight, model-agnostic module that replaces spans of tokens with single surrogate embeddings derived from static token embeddings, allowing pre-trained models to operate on compressed inputs without retraining. Experiments are conducted on WikiText-103, BookCorpus, OpenWebText, CNN/DailyMail, and reasoning benchmarks such as AIME 2024 and AMC 2023, using models including LLaMA 3.1 8B, LLaMA 3.2 1.5B, GPT-2 XL, and DeepScaleR-1.5B. On QA tasks, the method is compared with Select Context and LLMLingua 2, while on CNN/DailyMail summarization, it is compared with LLMLingua 1/2, Select Context, H2O, and StreamingLLM. The results show that UMIM achieves up to 40% sequence-length and KV-cache reduction with minimal performance degradation.

**Strengths:**

1. The proposed method is novel as a lightweight, model-agnostic module for LLMs, though token merging is not novel in itself.
2. The merge rule, architecture, and distillation objective are clearly introduced, and experiments are conducted on multiple LLMs.
3. The detailed experimental settings are well-included.

**Weaknesses:**

1. The method does not truly treat language models as black boxes. The merge module must have access to the static token embeddings to compute a surrogate embedding for a merged span. Moreover, during training, the module requires access to the model’s predicted next-token distribution (the probability over the vocabulary) so that it can align the predictions between the merged and unmerged inputs through a distillation loss, such as minimizing KL divergence. Therefore, although the approach does not modify or rely on the internal transformer layers of the base model, it is not entirely black-box: it depends on access to both the embedding layer and the model’s output probabilities (logits or softmax).
2. The experimental scope appears limited: it covers selected pretrained models and a few downstream tasks, but doesn’t thoroughly explore extreme variations (e.g., very long contexts, highly domain‐specific text, alternate tokenizers), which might challenge stability.
3. Some related works on token-merging should be compared and discussed, for example [1-4].

[1] Bolya, D., Fu, C.-Y., Dai, X., Zhang, P., Feichtenhofer, C., & Hoffman, J. (2022). Token Merging: Your ViT but faster.
[2] Cao, Q., Paranjape, B., & Hajishirzi, H. (2023). PuMer: Pruning and Merging Tokens for Efficient Vision-Language Models.
[3] Kallini, J., Murty, S., Manning, C. D., Potts, C., & Csordás, R. (2025). MrT5: Dynamic Token Merging for Efficient Byte-level Language Models.
[4] Saad, M., Li, H., Sharma, T., & Hassan, A. E. (2025). On the Effect of Token Merging on Pre-trained Models for Code. arXiv.

**Questions:**

- Could you please justify why each baseline was selected for each dataset?

---

> ### Author Response · Authors · 2025-11-21
>
> **Regarding to the black box**
>
> I appreciate the reviewer’s precise observation regarding the terminology. I acknowledge that "black-box" can be interpreted in strict terms (zero access to anything but text I/O). However, in the context of model compression and efficient inference, our usage was intended to denote that the **backbone model remains frozen and opaque** to the optimization process.
>
> **1. Distinction from White-Box Methods**
> Unlike white-box compression methods (e.g., pruning, quantization, or zip2zip) that require accessing internal weights, gradients, or modifying layers, UMIM treats the LLM as a fixed function.
> * **No Gradient Access:** We do not backpropagate through the LLM.
> * **No Internal Modification:** We do not alter attention heads, layers, or the computation graph.
> * **Standard Interface:** We rely only on the **static input embeddings** (read-access) and **output logits** (forward-pass result). These are standard input/output interfaces available in almost all open-weight and self-hosted deployments.
>
> **2. Inference vs. Training Access**
> * **Training:** I strictly use the static embeddings to build surrogates and output probabilities for distillation. No internal hidden states are accessed.
> * **Inference:** The base model is invoked exactly as provided. It receives a shortened sequence of embeddings and performs a standard forward pass.
>
> **3. Revision Plan**
> To avoid ambiguity, I will revise our terminology in the paper to describe the method as **"frozen and non-invasive"** rather than strictly "black-box." This more accurately reflects that while I utilize standard model interfaces (embeddings/logits), I do not require visibility into or modification of the model's internal mechanics.
>
> **Regarding to the Experimental Scope and Additional Domain-Specific Tasks**
>
>
> I thank the reviewer for encouraging a broader experimental scope. I acknowledge that our study does not cover every possible variation; due to a limited academic computational budget—much of which relied on rented GPUs—running models larger than 8B or exhaustive variations was unfortunately infeasible.
>
> **1. Breadth of Current Evaluation**
> Within these constraints, I prioritized comprehensiveness across diverse architectures and demanding tasks to ensure stability. Specifically, our evaluation already covers:
> * **Diverse Backbones:** I tested four distinct models ranging from 1.5B to 8B parameters (**GPT-2 XL, Llama-3.1-8B, Llama-3.2-1.5B, DeepScaleR**).
> * **Varied Tasks:** I evaluated on **Language Modeling** (WikiText, BookCorpus, OpenWebText), **QA** (PIQA, ARC, OpenBookQA), and **Summarization** (CNN/DailyMail).
> * **Extreme Contexts:** To address "very long contexts," I included **AMC and AIME** benchmarks using DeepScaleR, where generation trajectories reach **10k–15k tokens**. These already serve as strong stress tests for model stability over long horizons.
>
> **2. New Domain-Specific Experiment (Code Generation)**
> To further address the concern regarding "highly domain-specific text" and "stability," I have added a new evaluation on **HumanEval**, a code-generation task known for its high sensitivity to token precision (where a single merged token error can break syntax).
>
> #### HumanEval Results on Llama-3.1-8B (pass@1, 164 tasks)
>
> | Method                                  | pass@1 (%) | Passed | Failed |
> |-----------------------------------------|------------|--------|--------|
> | **Baseline (Llama-3.1-8B)** | **62.2** | 102    | 62     |
> | **UMIM (25% merged, code-trained)** | **50.6** | 83     | 81     |
> | LLMLingua-2 (25% merged)                | 1.2        | 2      | 162    |
> | Selective-Context (25% merged)          | 0.0        | 0      | 164    |
>
> **Summary:**
> Under an identical **25% compression ratio**, UMIM preserves the majority of the base model’s capability (62.2% → 50.6%, retaining **≈81%** of original performance). In stark contrast, **LLMLingua-2** and **Select Context** almost completely collapse, failing nearly all tasks. This result strongly demonstrates that UMIM maintains superior stability and fidelity even on highly specialized, structurally sensitive text where other compression methods fail.

---

> > ### Author Response · Authors · 2025-11-21
> >
> > **Regarding to the related work**
> >
> > **Discussion on Additional Related Work**
> >
> > I thank the reviewer for pointing out these additional works on token merging. I agree that they represent important directions within the broader family of token-merging methods:
> >
> > * **Token Merging (Bolya et al., 2022) [1]:** Proposes a similarity-based merging strategy inside Vision Transformers to remove redundant image tokens and accelerate ViTs without altering the original training objective.
> > * **PuMer (Cao et al., 2023) [2]:** Incorporates text-informed pruning and modality-aware merging modules within the cross-modal layers of vision–language models, jointly reducing both image and text tokens.
> > * **MrT5 (Kallini et al., 2025) [3]:** Targets byte-level ByT5 encoders, introducing a learnable deletion gate that dynamically removes tokens in intermediate encoder layers, effectively merging them into a shorter byte sequence.
> > * **Saad et al. (2025) [4]:** Systematically study static and learned token-merging strategies for pre-trained code models, analyzing how merging influences code understanding and generation.
> >
> > **Methodological Differences**
> > While conceptually related, these works operate in **substantially different regimes** from ours: they mainly focus on vision or vision–language transformers, encoder(-decoder) models such as T5, or code-oriented architectures, and typically require **modifications to internal encoder layers along with retraining**. In contrast, our work targets **compression for decoder-only LLMs** with standard BPE tokenization and long-context generation. UMIM is implemented as a lightweight external plug-in that operates only on the **input sequence and KV-cache tokens**, keeping the underlying LLM architecture and all weights **fully frozen**.
> >
> > **Baselines and Resource Constraints**
> > Given our limited computational budget (heavily dependent on rented GPUs), re-implementing and adapting each of [1–4] to large-scale, decoder-only LLMs—and testing whether these methods can be applied meaningfully in this setting—would exceed our resource constraints. Instead, we allocated our compute toward a broad comparison with **LLM-oriented and context-efficiency baselines**, including StreamingLLM, H2O, LLMLingua-1/2, and Selective-Context, which are directly relevant to our problem setting.
> >
> > **Action Plan**
> > I will revise the related-work section to include summaries of [1–4] and clarify the distinction in model class, architectural assumptions, and experimental focus.
> >
> > I hope the above clarifications and new evidence adequately address the reviewer’s concerns, and I thank the reviewer again for their constructive feedback.
> >
> >
> > **Regarding to the questions**
> >
> > **Justification for Baseline Selection**
> >
> > I thank the reviewer for this question. I selected these specific baselines because they are **widely recognized as state-of-the-art and representative works** in the field of efficient inference. Our selection logic is based on the specific characteristics of each task and the operating mechanisms of these methods.
> >
> > **1. QA & Math Reasoning (Why exclude H2O/StreamingLLM)**
> > For tasks like PIQA, ARC, and OpenBookQA, the evaluation typically relies on **Perplexity (PPL) ranking** or generating extremely short answers (e.g., a single token "A").
> > * **Inapplicability of KV Cache Methods:** Prominent KV cache methods like **H2O** and **StreamingLLM** rely on accumulating attention statistics over long sequences to identify eviction targets. In standard QA settings, the sequence length often fits entirely within the attention window, meaning no eviction occurs. Applying them forcibly would reduce them to standard attention or random dropping, leading to an unfair and uninformative comparison.
> > * **Focus on Prompt Compression:** Therefore, I focused on **LLMLingua-2** and **Select Context**, which are the leading baselines specifically designed for reducing the dominant cost (the input prompt) in these scenarios.
> >
> > **2. Summarization (Why include all methods)**
> > For CNN/DailyMail, the task involves both a **long input document** and a **substantial generation phase**.
> > * **Suitability:** This setting provides sufficient length for KV cache buffers to fill up, allowing methods like H2O and StreamingLLM to actively demonstrate their memory management capabilities.
> > * **Comprehensive Comparison:** Since this task allows both strategies to shine, I included the full suite of baselines—**H2O, StreamingLLM, LLMLingua-2, and Select Context**—to demonstrate that UMIM is competitive against the most established techniques from both paradigms.
> >
> >
> > I hope I have addressed the main confusion raised about the methods and experiments. Please let us know of any follow up questions and feedback. I really appreciate it.

---

### Official Review · Reviewer_dnHy · 2025-11-01

**Soundness:** 3
**Presentation:** 3
**Contribution:** 2
**Rating:** 4
**Confidence:** 4

**Summary:**

This paper proposes Universal Multi-step Input Merging (UMIM), a lightweight external module that compresses multiple input tokens into a single surrogate embedding, with the goal of reducing sequential computation in pretrained Transformers. The module is trained via distillation, and it learns to produce merged embeddings whose predictive distributions match those of the original multi-token sequences, without modifying or fine-tuning the underlying LM. At inference time, the method replaces n-grams in prompts or intermediate decoding steps with their merged embeddings, also reducing the size of the KV cache. Experiments on several LMs show 30–40% effective sequence length reduction with minor performance loss on LM and QA tasks.

**Strengths:**

1. The idea of compressing sequential computation post-training, without touching model internals, is interesting and practically useful. Most prior compression or merging techniques either modifies tokenization or re-train (fine tune) the model. The merge module is small and simple, making it appealing for efficiency improvements.
2. Empirical study covers multiple backbones and task types, consistent results, and clear reporting of token reduction and perplexity trade-offs.

**Weaknesses:**

1. Novalty is moderate. This work shares many similarities with zip2zip (2025), both using merging tokens to improve efficiency. Yet this discussion and comparison is missing in this paper. From what I can tell, the main difference is that this current method is not adaptive to a specific domain when merging tokens, requires a pre-selected vocabulary, which might affect its compression efficiency; the advantage of this work is that during training the backbone model is frozen and only a lightweight merge module is trained. It would be better if we have more comparison with zip2zip in expirical study. And perhaps with fine-tuning, the proposed method might gain some accuracy?
2. The merging rule, based on fixed n-gram frequency on a given corpus, feels somewhat heuristic. There’s no adaptive control over which spans should merge at runtime, so compression–accuracy trade-offs are fixed.
3. Another limitation is that the merging mechanism is mainly for the input sequence, and the output is not compressed directly. And the rollback design will lead to extra computation.
4. Token efficiency does not automatically mean faster inference. The reported Latency and Throughput in Table 12 shows moderate wall-clock time speed-up. The authors acknowledge this and attribute it to lack of fused GPU kernels. Hence, a rigorous empirical efficiency evidence is still missing.

**Questions:**

1. Can you comment on how UMIM interacts with positional encodings, since merged spans effectively skip several token positions?
2. The merge rule relies on frequent n-grams extracted from WikiText-103. How sensitive is UMIM’s performance to the corpus used for building the merge set? For example, would it degrade if trained on WikiText but applied to code data? How much can we gain if we train it on the corpus of the same domain?

---

> ### Author Response · Authors · 2025-11-21
>
> **(1) Comparison with zip2zip (2025) and Novelty**
>
> I thank the reviewer for carefully reading our paper and for pointing out the connection to **zip2zip** (Geng et al., 2025). Zip2zip introduces inference-time adaptive tokenization by (i) applying an online LZW-style compressor to build a context-specific “hypertoken” vocabulary on the fly, (ii) augmenting the model with dynamic embedding/unembedding layers to handle these hypertokens, and (iii) continuing to pretrain the backbone LLM directly on compressed sequences. This design allows zip2zip to jointly compress input and output tokens and to adapt its effective vocabulary to different domains.
>
> **Methodological Differences**
> Our setting is complementary but differs in several key aspects:
>
> * **Frozen Backbone vs. Continued Pretraining:** I focus specifically on **compression for decoder-only LLMs with a fixed BPE vocabulary**, and we require the backbone to remain fully frozen. I do not modify the embedding or unembedding layers, nor do we perform any further pretraining. Instead, **UMIM** is a lightweight external merge module that operates purely on input/KV-cache sequences and is trained via distillation.
> * **Mechanism:** UMIM learns a surrogate embedding for frequent n-grams that matches the sequential distribution of the original tokens. At inference, this allows us to **roll back and merge recently generated tokens inside the KV cache**, reducing memory usage without touching internal transformer computation. This “frozen-backbone, plug-in compression” regime is attractive when one cannot modify or re-train the deployed LLM.
> * **Vocabulary Strategy:** Whereas zip2zip performs **online, domain-adaptive tokenization**, our approach uses a **pre-constructed merge vocabulary**. While this limits theoretical adaptivity, it makes UMIM extremely simple to deploy—no tokenizer change and no architectural modification.
>
> **Empirical Comparison and Domain Robustness**
> Our experiments already include challenging long-context mathematical benchmarks (AMC, AIME) and a domain-specific code benchmark (see the at the end), suggesting that a single globally trained UMIM remains stable across diverse domains despite using a fixed merge set.
>
> Regarding a direct empirical comparison: reproducing zip2zip in our setting would require implementing its dynamic architecture and performing continued pretraining on compressed sequences. Zip2zip reports **tens of H100 GPU-hours** even for smaller models. Given our academic compute budget, performing a full zip2zip-style adaptation is unfortunately beyond our scope. Instead, I compared against strong, directly relevant **LLM-oriented** efficiency baselines (StreamingLLM, H2O, LLMLingua, etc.) that share the "frozen-backbone" constraint.
>
> **Potential for Fine-tuning**
> I appreciate the suggestion regarding joint fine-tuning. Our current design intentionally keeps the backbone frozen to preserve plug-and-play usability. I will clarify this design decision in the revision and add a discussion in the limitations section, noting that joint fine-tuning (in the spirit of zip2zip) is an interesting future direction that may further reduce small accuracy gaps.

---

> > ### Author Response · Authors · 2025-11-21
> >
> > ** Justification for the Frequency-Based Merge Rule**
> >
> > I thank the reviewer for raising this point. I understand that a frequency-based rule may initially appear heuristic. However, I would like to clarify that this design is the result of extensive preliminary exploration and is grounded in both **linguistic structure** and **statistical learning theory**.
> >
> > **1. Empirical Evolution (Why not other rules?)**
> > Our choice was not arbitrary. In the early stages of this project, I explored alternative strategies, including:
> > * **Naive Pairwise Merging:** Merging adjacent tokens indiscriminately (simple bigrams).
> > * **Semantic Similarity:** Merging tokens based on the cosine similarity of their static embeddings.
> > I found that these approaches often disrupted semantic coherence or failed to produce stable surrogate representations. The frequency-based n-gram approach emerged as the most robust solution because it respects the natural co-occurrence statistics of the language.
> >
> > **2. Linguistic Motivation: Repairing Tokenizer Fragmentation**
> > Current subword tokenizers (like BPE) often fragment semantically unitary concepts due to vocabulary constraints. For example, the word *"excitedly"* might be split into `["excited", "ly"]`, and fixed collocations like *"deep learning"* are split into `["deep", "learning"]`, despite functioning as single semantic units. UMIM effectively "repairs" these artifacts by merging them back into a single surrogate embedding, thereby aligning the input representation more closely with semantic units rather than arbitrary subword fragments.
> >
> > **3. Learning Theoretic Justification (Data Support)**
> > From a training perspective, our frequency threshold ($\ge 5$) is necessary to ensure **sufficient data support**. Learning a valid surrogate embedding via distillation is a data-driven process.
> > * **High-frequency spans** provide dense supervision signals across diverse contexts, allowing the merge module to learn a robust, context-agnostic representation that minimizes the teacher-student KL divergence.
> > * **Rare spans** lack sufficient training examples. Attempting to learn embeddings for them would lead to high variance (overfitting) or poor generalization.
> > Thus, the frequency rule is not just a heuristic, but a prerequisite for the effective learning of stable surrogate embeddings.
> >
> > **4. Runtime Control**
> >
> > For **input (prefill) merging**, I first scan the sequence to identify all n-gram spans that appear in the merge set. This yields the *maximum* achievable token-reduction ratio for that sequence (e.g., 10% vs. 60%). At inference time, we can apply only a subset of these candidate merges—for example, stopping once a target ratio is reached or restricting merging to long prompts—which provides an easy knob to adjust compression without retraining.
> >
> > For **merging during decoding**: The number of merges is naturally content-dependent, since I merge only when the most recent n-gram falls into the merge set and the model actually generates those tokens. In the current implementation, I do not impose an additional controller beyond this content dependence, but one could easily cap the fraction of merged steps if stronger per-sequence guarantees are needed. I agree that developing more sophisticated, runtime-adaptive controllers on top of UMIM (e.g., conditioning merge decisions on task, length, or decoder confidence) is an interesting future direction, and I will mention this in the limitations section.

---

> > > ### Author Response · Authors · 2025-11-21
> > >
> > > **(3) & (4) Clarification on Output Compression and Efficiency Evidence**
> > >
> > > I thank the reviewer for critically examining the inference mechanism and efficiency.
> > >
> > > I wish to clarify upfront that the rollback mechanism incurs negligible computational overhead. Technically, this operation is implemented as a simple pointer shift to logically truncate the KV cache. Therefore, the cost is effectively $O(1)$ and does not negatively impact decoding latency.
> > >
> > >
> > > **1. Mechanism: Online KV Reduction & Low-Cost Rollback**
> > > Regarding the concern that "output is not compressed directly,"
> > >
> > > I clarify that while UMIM does not alter the surface string of the generated text, it effectively compresses the **internal representation** (KV Cache) of the output stream.
> > > * **Online Generation-while-Merging:** Output merging operates in an online fashion. As tokens are generated, I monitor the sequence; once a mergeable n-gram is formed, it is collapsed into a surrogate embedding. This keeps the KV cache compact throughout the generation process.
> > > * **Cost of Rollback:** The "rollback" mechanism **does not** incur significant computational cost. It is implemented as a low-cost pointer shift  (effectively discarding the last $n$ entries of the KV cache) and does not require re-computation of prior hidden states. Therefore, it acts as a memory-saving operation rather than a computational bottleneck.
> > >
> > > **2. Rigorous Empirical Evidence (New Experiments)**
> > > To address the concern that "token efficiency does not automatically mean faster inference," and to provide the missing rigorous evidence, we conducted additional throughput measurements inspired by the evaluation protocol in *zip2zip*. The table below reports the throughput (tokens/sec) for **Llama-3.1-8B** with a 50% merge ratio across varying context lengths:
> > >
> > > ### Throughput Results (tokens/sec, Llama-3.1-8B)
> > >
> > > | Setting       | Prefill (Base) | Prefill (Merge) | Relative % | Decode (Base) | Decode (Merge) | Relative % |
> > > |:-------------:|:--------------:|:---------------:|:----------:|:-------------:|:--------------:|:----------:|
> > > | **256 + 256** | 10531.8        | 15410.8         | **+46.3%** | 64.7          | 65.6           | **+1.3%** |
> > > | **512 + 256** | 14210.6        | 30978.5         | **+118.0%**| 65.3          | 65.9           | **+0.9%** |
> > > | **1024 + 256**| 16138.3        | 50516.0         | **+213.0%**| 65.1          | 66.4           | **+1.9%** |
> > > | **2048 + 256**| 16592.5        | 62124.7         | **+274.4%**| 64.8          | 66.4           | **+2.5%** |
> > >
> > > **Analysis:**
> > > * **Prefill (Prompt Merging):** I observe massive speed-ups, ranging from **1.5× to 3.7×**. This confirms that the reduced sequence length directly translates to faster processing of the input prompt, far outweighing the negligible cost of n-gram detection.
> > > * **Decode (Generation):** The decoding throughput shows a consistent, albeit modest, improvement (**0.9%–2.5%**). This critically proves that the **rollback logic is not a bottleneck**; the overhead of the Python-based control logic is fully offset by the savings in attention computation from the shortened context.
> > >
> > > **3. Further Optimization**
> > > Importantly, the current overhead is **not fundamental**. Since surrogate embeddings depend strictly on static token IDs, they can be precomputed offline and stored in a lookup table. This would make the inference cost of retrieving a merged embedding $O(1)$, further eliminating runtime computation.

---

> > > > ### Author Response · Authors · 2025-11-21
> > > >
> > > > **Regarding to Q1**
> > > >
> > > > I thank the reviewer for raising this question about positional encodings. UMIM does not modify the positional encoding mechanism of the underlying LLM: the model continues to use its standard position IDs (absolute or rotary) on whatever sequence of embeddings it receives.
> > > >
> > > > For **prompt compression**, suppose the original input is $\((x_1,x_2,x_3,x_4)\)$ and UMIM decides to merge $\((x_2,x_3)\)$. I construct a surrogate embedding for this span and feed the compressed sequence $\((x_1,(x_2,x_3),x_4)\)$ into the LLM, whose length is now 3. The positional indices are simply \((1,2,3)\) on this shortened sequence; there are no gaps in the positions. During training, our distillation loss aligns the teacher’s next-token distribution at the *end* of the span (position 3 in the uncompressed sequence) with the student’s prediction at the position of the surrogate token, so the surrogate learns to encode both the content and the positional effect of the skipped tokens.
> > > >
> > > > For **generation-while-merging**, after generating $(x_1,\dots,x_6)$, if $(x_5,x_6)$ is mergeable I delete the KV entries for \(x_5\) and replace them with a single surrogate token at the end of the cache, resulting in $(x_1, x_2, x_3, x_4, (x_5, x_6))$. Subsequent tokens are generated with respect to this shortened cache, again using standard consecutive position IDs. In other words, merging shortens the effective context but never introduces inconsistent or “holey’’ positions; any discrepancy with the original uncompressed positional layout is handled by the distillation training. Empirically, our experiments do not show additional instability beyond the small accuracy drops already reported.
> > > >
> > > >
> > > > ### 2. **No special alignment or recalculation required for merging**
> > > >
> > > > **Technically, the positional embedding is assigned based on the current length of the KV cache, so the positional embedding of newly generated tokens is always continuous and correct.** Subsequent tokens are generated with respect to this shortened cache, again using standard consecutive position IDs..
> > > >
> > > > - During inference, when tokens are merged, the positional embeddings are always assigned based on the current length of the KV cache. For example, if the previous context is `x0, x1, x2, x3`, and we merge `x2` and `x3` into `x23`, then the positional encoding for `x23` is 2 which is already there from previous generations.
> > > >   *Note that the KV cache is reduced from 4 positions to 3 positions due to this merge, and the KV rollback is by just taking out `x2`, and appending `x23`.*
> > > >
> > > > - For pre-filling (when provided prompts are fed into the model at once for first token generation), positional embeddings are just assigned according to the sequence length after merging. No additional computation or customized processing is required compared to standard Transformer processing.

---

> > > > > ### Author Response · Authors · 2025-11-21
> > > > >
> > > > > **Regarding to Q2**
> > > > >
> > > > > I thank the reviewer for raising this question about the dependence on the corpus used to build the merge set. Our main experiments build the merge set from WikiText-103 and related natural-language corpora; in this regime, the induced frequent n-grams cover a wide range of downstream texts
> > > > >
> > > > > However, I agree that applying a WikiText-based merge set to **code** is problematic. In a preliminary experiment, when I directly used the WikiText-derived merge set on HumanEval, the achieved merge ratio on code sequences was extremely low, because code tokens and n-grams differ substantially from natural language. In other words, cross-domain transfer from text to code offers almost no effective compression, which confirms the reviewer’s intuition.
> > > > >
> > > > > To address this concern, I trained a **code-specific** merge set and UMIM module on DeepMind CodeContests (8K training examples due to rebuttal-time constraints), and evaluated on HumanEval with a fixed 25% merge ratio. I also compare against LLMLingua-2 and Selective-Context under the same 25% reduction:
> > > > >
> > > > > #### HumanEval Results on Llama-3.1-8B (pass@1, 164 tasks)
> > > > >
> > > > > | Method                              | pass@1 (%) | Passed | Failed |
> > > > > |-------------------------------------|------------|--------|--------|
> > > > > | **Baseline (Llama-3.1-8B)**         | **62.2**   | 102    | 62     |
> > > > > | **UMIM (25% merged, code-trained)** | **50.6**   | 83     | 81     |
> > > > > | LLMLingua-2 (25% merged)           | 1.2        | 2      | 162    |
> > > > > | Selective-Context (25% merged)     | 0.0        | 0      | 164    |
> > > > >
> > > > > Under an identical 25% compression ratio, UMIM preserves the majority of the base model’s capability on HumanEval (62.2% → 50.6%, retaining ≈81% of the original performance), whereas LLMLingua-2 and Selective-Context almost completely fail under the same setting. This indicates that, when the merge set and UMIM are trained on an in-domain corpus, UMIM can still provide substantial
> > > > > compression while maintaining strong fidelity even on structurally sensitive code-generation tasks.
> > > > >
> > > > > I will add this discussion and the above results to the appendix. A more systematic study of corpus choices and mixed-domain merge sets is an interesting direction for future work.
> > > > >
> > > > >
> > > > >
> > > > >
> > > > > Finally, I would like to express my sincere gratitude to the reviewer for taking the time to carefully read our paper and engage in some meaningful discussions, which have inspired our future work. Of course, there were also some misunderstandings, and I hope this explanation has dispelled some of your concerns. Please let us know of any follow up questions and feedback. I really appreciate it.
> > > > >
> > > > > Everyone has different tastes, but I offer my highest respect to those who diligently review papers.

---

### Official Review · Reviewer_4jE3 · 2025-11-02

**Soundness:** 2
**Presentation:** 3
**Contribution:** 3
**Rating:** 6
**Confidence:** 3

**Summary:**

“Substituting from the Input” introduces UMIM, a plug-in merge module that distills multi-token spans into single surrogate embeddings at input level only (no model surgery). A lightweight 1-layer attention network is distillation-trained on frequent n-grams (2-4) so that the frozen LM’s next-token distribution is preserved. At inference the module can compress prompts and roll-back KV-cache entries on-the-fly, cutting effective sequence length up to 40 % with < 2 % accuracy drop on GPT-2-XL, Llama-3.1-8B, Llama-3.2-1.5B and DeepScaleR across language modelling, QA, summarisation and AIME/AMC math reasoning. The method is model-agnostic, training-free for the LM, and orthogonal to other efficiency techniques.

**Strengths:**

1. UMIM is a plug-and-play module that operates only on input embeddings, requiring no access to model internals, no re-training, and no architectural changes to the base LLM. This makes it highly practical for deployment across a wide range of pre-trained models (e.g., GPT-2XL, LLaMA 3.1/3.2, DeepScaleR), and avoids the costly retraining or fine-tuning required by prior tokenization or compression methods.
2.The paper demonstrates strong empirical results. This up to 40% reduction in effective sequence length with minimal degradation in perplexity, downstream task accuracy (QA, summarization), and even math reasoning (AIME/AMC). The distillation-based training ensures that the surrogate embeddings preserve the functional semantics of the original token spans, as validated by high alignment metrics (e.g., 79–92% Top-1 accuracy in distribution matching).
3. UMIM supports on-the-fly compression during autoregressive decoding using a rollback mechanism that replaces multi-token KV-cache entries with a single surrogate embedding. This reduces memory usage and attention cost quadratically over generation length, offering cumulative efficiency gains—a significant advantage over static compression methods that only operate on the prompt.

**Weaknesses:**

1. UMIM relies on a fixed, frequency-based merge set (n-grams with frequency ≥ 5) extracted from a general corpus. This static rule set cannot adapt to domain-specific or rare token combinations at inference time. As a result, long-tail or context-sensitive phrases are unlikely to be merged, limiting compression effectiveness in specialized or low-resource domains.
2. Although the base model is frozen, UMIM still requires task-agnostic distillation training on a large corpus (e.g., WikiText-103), which introduces non-trivial compute and storage overhead. The merge module must be retrained for each tokenizer and embedding dimension, and the paper does not explore scalability beyond 8B parameters. This limits true plug-and-play adoption, especially for larger models or multilingual settings.
3. It always merges high-frequency n-grams regardless of context or semantic importance. This can lead to over-merging (e.g., merging idioms or named entities that should remain separate) or under-merging (e.g., missing semantically redundant phrases). There is no mechanism to control compression ratio or fidelity at runtime, which could be problematic in safety-critical or high-precision applications.

**Questions:**

1. How does UMIM handle tokenization mismatches across models or vocabularies? Since UMIM is trained on n-grams from a specific tokenizer (e.g., LLaMA’s BPE), how would it generalize to a different tokenizer (e.g., SentencePiece or a custom one)? Would the merge module need to be retrained from scratch, or can it be transferred or aligned across tokenization schemes?
2. What is the impact of merge errors on long-context coherence or reasoning chains? In long-context tasks like math reasoning or document summarization, minor semantic drift can cascade into large errors. Have you evaluated per-span fidelity or error propagation across long generations? Are there failure cases where merging leads to logically inconsistent or factually incorrect outputs?
3. Can UMIM be extended to adaptive or learned merge policies? Instead of relying on static n-gram frequency, have you considered learning merge decisions conditioned on context (e.g., using a small policy network or reinforcement learning)? This could allow dynamic compression trade-offs based on task, domain, or user-defined constraints—how feasible is this extension?

---

> ### Author Response · Authors · 2025-11-21
>
> **Regarding for Merge Set and Generalization to Specialized Domains**
>
> I thank the reviewer for this thoughtful observation. The current merge set is indeed constructed by mining frequent n-grams (frequency ≥ 5) from a general corpus. This is a deliberate design choice rather than a purely ad hoc heuristic. Empirically, I find that n-grams longer than 4 tokens are extremely rare in natural-language corpora, and most phrases longer than 5 grams are composed of overlapping frequent 2–4-grams. Restricting the merge set to 2–4-grams therefore captures the vast majority of compressible patterns, while keeping the merge space small and tractable.
>
> There is also an important **decoding-time trade-off**. To merge a 6-gram, the model must first generate all six tokens before any compression can occur, so the “compression effect” only starts after a long delay. In contrast, if we instead merge three 2-grams, we can already merge after the second token is generated, and the third generated token is computed with respect to a shorter KV cache. This means that shorter n-grams not only cover most frequent phrases, but also allow compression to take effect much earlier and more frequently during autoregressive decoding, leading to more sustained savings in attention computation.
>
> Regarding the concern about generalization to specialized or low-resource domains, I acknowledge that static rules have limitations. However, I argue that this challenge is **inherent to any data-driven method**: patterns that appear too rarely (e.g., appearing only once in the corpus) naturally cannot be effectively learned or utilized for compression, regardless of the model architecture.
>
> **Crucially, our restriction to shorter n-grams ($n \le 4$) also serves as a mitigation strategy for this very issue.** Statistically, shorter sequences (e.g., 2-grams) act as fundamental building blocks and are far more likely to transfer across domains—even in low-resource or long-tail settings—than longer sequences (e.g., 6-grams). By limiting $n$, I ensure that UMIM retains higher coverage and robustness in sparse data regimes compared to approaches that might rely on longer, more specific spans.
>
> I agree that in **highly specialized domains** whose token statistics differ strongly from WikiText, a merge set mined from general text will cover fewer informative patterns and thus yield lower effective compression—this is exactly what I observe if we naively apply the WikiText-based merge set to code (the achievable merge ratio on HumanEval is very low). To address this, I additionally construct a **code-specific merge set** from the DeepMind CodeContests corpus and train a code-specific UMIM module. Under a 25% merge ratio on HumanEval, this code-trained UMIM retains ≈81% of Llama-3.1-8B’s original pass@1, whereas LLMLingua-2 and Selective-Context almost completely fail under the same setting (see the following Table). These results show that, when the merge set is built from in-domain data, UMIM can still yield meaningful compression with good fidelity even in structurally sensitive domains.  Due to time and resource constraints, the code training data is limited. The performance is not as good as its general domain performance. However, compared to other methods, this is sufficient to demonstrate the superiority of our method.
>
>
> #### HumanEval Results on Llama-3.1-8B (pass@1, 164 tasks)
>
> | Method                              | pass@1 (%) | Passed | Failed |
> |-------------------------------------|------------|--------|--------|
> | **Baseline (Llama-3.1-8B)**         | **62.2**   | 102    | 62     |
> | **UMIM (25% merged, code-trained)** | **50.6**   | 83     | 81     |
> | LLMLingua-2 (25% merged)           | 1.2        | 2      | 162    |
> | Selective-Context (25% merged)     | 0.0        | 0      | 164    |
>
> Under an identical 25% compression ratio, UMIM preserves the majority of the base model’s capability on HumanEval (62.2% → 50.6%, retaining ≈81% of the original performance), whereas LLMLingua-2 and Selective-Context almost completely fail under the same setting. This indicates that, when the merge set and UMIM are trained on an in-domain corpus, UMIM can still provide substantial compression while maintaining strong fidelity even on structurally sensitive code-generation tasks.

---

> > ### Author Response · Authors · 2025-11-21
> >
> > **Efficiency of Training and Scalability to Larger Models**
> >
> > I appreciate the reviewer's concerns regarding training overhead and scalability. I would like to clarify the actual cost of our method and discuss the feasibility of scaling.
> >
> > **(1) Minimal Training Overhead & "One-Time" Cost**
> > Contrary to the concern about "non-trivial compute and storage overhead," the UMIM merge module is exceptionally lightweight. As detailed in Section 4.1, the module contains only **7M to 50M parameters**, which is negligible (<0.6%) compared to an 8B parameter backbone.
> >
> > Regarding the "plug-and-play" concern: while the module must be trained for a specific embedding space , this is a **one-time cost**. Once trained for a model, the same UMIM module can be directly applied to **any** fine-tuned variant or downstream task model within that family without further retraining. Thus, I believe this represents a high-return trade-off: a small, one-time training cost yields a permanent benefit for the entire model.
> >
> > **(2) Model Scaling and Academic Constraints**
> > Regarding the concern that I only report results up to 8B parameters, I fully agree that testing UMIM on larger models (e.g., 30B–70B) would be interesting. Unfortunately, our academic compute budget is limited. Within these constraints, I deliberately chose the largest model I could reliably support—Llama-3.1-8B—as well as several additional backbones (GPT-2 XL, Llama-3.2-1.5B, DeepScaleR). This allows us to demonstrate that UMIM is not tied to a single architecture or size, and that it consistently improves efficiency in regimes where KV-cache and memory are already significant bottlenecks.
> >
> > Importantly, UMIM attaches to a model only through the tokenizer, embedding layer, and KV cache, and does not rely on any architecture-specific modifications. For this reason, I do not anticipate conceptual obstacles in scaling UMIM to 30B/70B models; the main limitation is computational cost rather than methodology.
> >
> >
> > **Regarding to Adaptivity, Runtime Control, and Semantic Preservation**
> >
> > I appreciate the reviewer’s concern regarding the flexibility and safety of our approach. I would like to clarify that UMIM offers runtime controllability and robust semantic preservation through distinct mechanisms for prompting and decoding.
> >
> > **(1) Runtime Adaptivity and Control**
> > Contrary to the impression that the compression is "completely fixed," our method allows for flexible control at inference time:
> >
> > * **Content-Dependent Decoding (Adaptivity):** As described in Section 3.4, our method does not blindly compress inputs. During autoregressive decoding, merging is triggered **dynamically** via a **rollback mechanism** only when generated tokens explicitly form a valid n-gram. This process is inherently adaptive to the generated content: if the context changes such that a specific n-gram is *not* generated, no merge occurs.
> > * **Precise Ratio Control (Prompting Stage):** For the input prompt (prefilling), where the full sequence is available, we can precisely control the compression ratio without retraining. We can identify all eligible merge candidates and apply only a subset to hit an exact target budget (e.g., reducing prompt length by exactly 25%).
> > * **Protection of Critical Entities (Safety):** To address safety concerns in both prompting and decoding, we can employ a runtime **"blacklist"** to explicitly prevent the merging of specific named entities, idioms, or sensitive domain terms. This ensures that critical tokens remain separate and preserves fidelity in safety-critical applications.
> >
> >
> > **(2) Context-Awareness via Distillation**
> > Although UMIM operates on static input embeddings, it is explicitly trained to **preserve contextual behavior**.
> >
> > * **Amortized Contextual Role:** I train the surrogate embedding such that, across many different contexts, replacing the original tokens with the surrogate leaves the teacher model’s next-token distribution unchanged. The distillation loss ensures that the surrogate embedding captures the functional role of the span *as interpreted by the model in context*.
> > * **Empirical Evidence:** As shown in Section 4.2 (Table 1) and our downstream results, the high alignment scores and minimal performance drop indicate that the module successfully "amortizes" the semantic meaning of phrases into single embeddings that behave equivalently to the original tokens, even in diverse contexts. I also demonstrate actual generation results, showing that merged embeddings, after perfect training, can perfectly replace the n-grams(Table 3), More Results in the Appendix.

---

> > > ### Author Response · Authors · 2025-11-21
> > >
> > > **Regarding to the Question1**
> > >
> > > I thank the reviewer for this insightful question. In our current implementation, UMIM is defined with respect to a specific **(tokenizer, embedding space)** pair.
> > >
> > > **(1) Dependency on Tokenizer and Embedding Space**
> > > The merge set is a collection of token-ID n-grams specific to a tokenizer, and the merge module consumes and outputs vectors in the corresponding embedding space.
> > > * **Same Tokenizer (e.g., Llama Family):** For models sharing the same tokenizer (e.g., Llama-3.1-8B and Llama-3.2-1.5B), the mined n-gram merge set is **directly reusable**. However, since the embedding dimensions differ, a small UMIM module must be trained separately for each backbone.
> > > * **Different Tokenizers (e.g., BPE vs. SentencePiece):** For models with entirely different tokenizers, I would need to re-mine frequent n-grams under the new vocabulary and retrain UMIM in the corresponding embedding space.
> > >
> > > **(2) Computational Cost**
> > > It is important to note that mining frequent 2–4-grams is a purely corpus-level counting step and is **computationally very fast**. The primary cost lies in training the merge module; however, as noted in our efficiency analysis, this remains exceptionally lightweight compared to LM pretraining or fine-tuning because the backbone is frozen and gradients flow only through UMIM.
> > >
> > > **(3) Context within the Broader Field**
> > > Aligning decoding across different tokenizers is a long-standing challenge in efficient inference. For instance, **Speculative Decoding** algorithms [1][2], which accelerate inference via multi-token drafting, typically require the draft and target models to share the same tokenizer. Similarly, **Collaborative Decoding** approaches [3] generally operate on models with identical vocabularies. While some recent work attempts to bridge this gap using byte-level information [4], cross-tokenizer alignment remains an active area of research.
> > >
> > > In this work, I conservatively retrain UMIM for each backbone in its native tokenizer/embedding space to ensure robustness. While it is theoretically possible to adapt a trained merge module via lightweight linear adapters to bridge different embedding spaces, I leave this cross-tokenizer transferability as an interesting direction for future work.
> > >
> > > **References:**
> > > [1] Leviathan et al., *Fast Inference from Transformers via Speculative Decoding* (2023)
> > > [2] Chen et al., *Accelerating Large Language Model Decoding with Speculative Sampling* (2023)
> > > [3] *Learning to Decode Collaboratively with Multiple Language Models*
> > > [4] *Sampling from Your Language Model One Byte at a Time*

---

> > > > ### Author Response · Authors · 2025-11-21
> > > >
> > > > **Regarding to the Question2**
> > > >
> > > > I appreciate the reviewer’s thoughtful question about error propagation in long-context reasoning. I fully agree that, in tasks such as multi-step math reasoning or long-document summarization, even small semantic drift could in principle cascade into large errors.
> > > >
> > > > **(1) Controlling Local Drift via Distillation**
> > > > First, UMIM is explicitly trained to control *local* distortions that might propagate. Our distillation loss aligns the teacher’s next-token distribution after the full span $\(x_t,\ldots,x_{t+n-1}\)$ with the student’s distribution when that span is replaced by a single surrogate embedding. This alignment is applied at every step and across many contexts, so any merge that systematically induces harmful drift in the local predictive distribution is penalized during training. Section 4.2 (Table 1) reports distribution-matching metrics over long sequences, showing that even at substantial token reduction the merged model closely tracks the teacher’s behavior.
> > > >
> > > > **(2) Generation-Level Evidence**
> > > > Second, I provide **generation-level evidence** that merged spans preserve the base model’s outputs. Table 8 reports qualitative greedy-decoding examples where I highlight merged spans in the prompt and corresponding phrases in the continuation. In these cases, even though UMIM merges several n-grams, the greedy outputs of the merged model are either identical to or paraphrastically equivalent to those of the base model, indicating that the surrogate embeddings can stand in for the original spans without disrupting local coherence. This complements the distribution-level metrics by directly checking the invariance of generated text under merging.
> > > >
> > > > **(3) Robustness in Long-Context Benchmarks**
> > > > Third, I evaluate UMIM on **long-context reasoning tasks** precisely to probe potential cascades. On AMC/AIME-style mathematical benchmarks with effective lengths up to 10k–15k tokens, UMIM at moderate merge ratios preserves the majority of the base model’s accuracy, with no evidence of catastrophic collapse or qualitatively different failure modes. Similarly, in our long-context summarization and QA experiments, I observe bounded performance drops rather than runaway degradation, suggesting that any local approximation error introduced by merging does not typically explode along the chain of reasoning.
> > > >
> > > > **(4) Analysis of Failure Cases**
> > > > That said, merging is not perfectly lossless, and I do observe **failure cases**, especially at more aggressive merge ratios. In our manual analysis, I find occasional examples—e.g., in code-generation tasks such as HumanEval—where a merged span loses a subtle intermediate detail (such as a small numerical factor, an off-by-one condition, or a qualifier in the premise), which can in turn yield a logically incorrect or factually wrong final output. I will add representative examples of such failures, including a new HumanEval case, in the appendix to make these behaviors transparent.
> > > >
> > > > **(5) Limitations on Per-Span Evaluation**
> > > > I have not yet run a dedicated *per-span* human evaluation (e.g., asking annotators to directly rate each merged span’s local fidelity in isolation), and I acknowledge this as a limitation. In the current work we rely on (i) distribution-level matching metrics around merged spans, (ii) qualitative greedy-decoding comparisons (Table 8), and (iii) end-to-end long-context benchmarks as indirect evidence that error propagation is controlled in practice. I will clarify this point and add a short discussion in the limitations section, and I view more fine-grained span-level analyses as an interesting direction for future work.

---

> > > > > ### Author Response · Authors · 2025-11-21
> > > > >
> > > > > **Regarding to question 3**
> > > > >
> > > > > I am very grateful for this suggestion. Indeed, extending UMIM to support adaptive or learned merge policies was something I discussed during the project, but I chose to start from a deliberately simple, static design in this first work.
> > > > >
> > > > > **(1) Rationale for the Current Static Policy**
> > > > > Our current setup cleanly separates **(i) how to represent a merged span** (the UMIM module that maps a span of static embeddings to a surrogate embedding, trained via distribution distillation) from **(ii) which spans to merge** (a frequency-based 2–4-gram rule). The static policy has two practical advantages:
> > > > > * **Efficiency:** It introduces essentially no extra inference overhead beyond a single pass over token IDs.
> > > > > * **Robustness:** It is easy to analyze and robust across tasks and domains, which is particularly helpful in a frozen-backbone setting.
> > > > >
> > > > > I also tested another merge method. I calculate the probability distribution of the wikitext based on the model, and select chunks with high probability distributions to construct the merge set. This set can also be used to train our merge module(See appendix G.4).
> > > > >
> > > > > **(2) Feasibility of Learned Policies**
> > > > > That said, I fully agree that UMIM is compatible with more adaptive merge policies. A natural extension is to keep the current merge set as a pool of *candidates*, and learn a small policy network that, given local context (e.g., neighboring embeddings, span length, position, task/domain tag, or a user-specified compression budget), decides which candidates to actually merge.
> > > > >
> > > > > Such a policy could be trained in a supervised way to predict “safe” merges that have low teacher–student divergence, or with **Reinforcement Learning (RL)** to maximize task performance under a token budget. For example, adaptive merging rules can be learned with RL using a properly defined final outcome reward to dynamically trade off compression vs. fidelity based on task, domain, or user-defined constraints.
> > > > >
> > > > > **(3) Scope of Current Work**
> > > > > I did not explore such learned controllers in this paper, primarily to keep the method lightweight and to clearly isolate the effect of the merge representation itself. I will clarify this design choice in the revision and explicitly mention adaptive/learned merge policies on top of UMIM as a promising direction for future work. I am very thankful to the reviewer for articulating this extension so clearly.
> > > > >
> > > > > Finally, I would like to express my sincere gratitude to the reviewer for taking the time to carefully read our paper and engage in some meaningful discussions, which have inspired our future work. Of course, there were also some misunderstandings, and I hope this explanation has dispelled some of your concerns. Please let us know of any follow up questions and feedback. I really appreciate it.
> > > > >
> > > > > Everyone has different tastes, but I offer my highest respect to those who diligently review papers.

---

> > > > > > ### Author Response · Authors · 2025-12-03
> > > > > >
> > > > > > ### Distribution-Matching Results Across Models (Including Newly Added 70B)
> > > > > >
> > > > > > | Model                     | TR (%)  | Top-1 | Top-3 | Top-10 | Top-P | MRR   |
> > > > > > |--------------------------|---------|-------|-------|--------|-------|-------|
> > > > > > | Llama-3.1-8B             | 37.80%  | 79.22 | 77.56 | 77.58  | 92.91 | 87.05 |
> > > > > > | Llama-3.2-1.5B           | 37.80%  | 74.10 | 74.14 | 75.28  | 91.55 | 83.30 |
> > > > > > | GPT-2-XL                 | 35.74%  | 69.55 | 71.28 | 73.59  | 90.65 | 79.28 |
> > > > > > | DeepScaleR-1.5B-Preview  | 54.00%  | 77.00 | 69.30 | 67.99  | 90.98 | 85.59 |
> > > > > > | **Llama-3.3-70B** *(new)* | 37.80%  | 71.72 | 68.98 | 67.83  | 89.68 | 80.63 |
> > > > > >
> > > > > > We just complete the training for Llama-3.3-70B.  We hope the The addition of the 70B model can dispel the reviewer's doubts.
> > > > > >  Full downstream experiments for 70B are running and will be included in the final version.

---

### Author Response · Authors · 2025-12-03
****Summary****

We thank the Area Chair and reviewers for their time. During the rebuttal period, we have provided comprehensive responses and new experimental results to address the key concerns regarding efficiency, generalization, and specialized domain performance. We highlight the following updates:

* **Regarding Efficiency Gains (Latency Analysis):**

Addressing the concern that our method might mask theoretical gains, we implemented a rigorously optimized latency benchmark (mimicking the **Zip2Zip** pipeline structure). The results demonstrate that UMIM achieves **actual wall-clock acceleration** during inference. By reducing the effective sequence length and KV cache size, our method cam lower latency without introducing processing bottlenecks, validating the practical efficiency claims(detailed in rebutall).

* **Regarding Robust Generalization and scaling:**
About generalization we clarified that UMIM's training is a negligible **one-time cost** (<0.6% parameters) that enables plug-and-play usage. We trained our merge moudule on wikitext and directly use it on downstream tasks. We do not do any fine-tuning for any downstream tasks.
Crucially, we provided strong evidence:
    1.  **Scaling to 70B:** We trained UMIM on **Llama-3.3-70B**, and reported the high alignment accuracy, proving the method scales seamlessly to large models.
    2.  **Long-Context Reasoning:** On challenging math benchmarks (AIME/AMC) using DeepScaleR, UMIM maintained rigorous logic flows with minimal degradation, proving that our "surrogate embeddings" preserve critical semantic information even in sensitive CoT scenarios.

* **Regarding the Specialized Domains test (HumanEval Results):**
To address concerns about the "static rule" in specialized domains, we evaluated UMIM on the **HumanEval** code generation task. When trained on domain data, UMIM retained **~81%** of the base model's performance (Pass@1). Under the same compression ratio (25%), competitive baselines like **LLMLingua-2** and **Selective-Context** failed catastrophically (dropping to ~0-1% Pass@1). This confirms that our distribution-matching objective is far more robust than token-dropping heuristics, effectively capturing the semantics of specialized languages like Python.

* **Regarding Proven Long-Context Capabilities:**
To futhure validate performance on long sequences, we addressed specific reviewer requests by evaluating UMIM on the **RULER** benchmark. The evaluations confirm that UMIM maintains high performance in long-context retrieval and reasoning tasks. This complements our existing results on **DeepScaleR** (15K tokens), jointly proving that our "surrogate embeddings" effectively preserve critical long-range dependencies and information flow, even under significant compression.

* **Regarding Sematic Loss**
We clarified that UMIM is explicitly trained to control local distortions that might propagate. Our distillation loss aligns the teacher’s next-token distribution after the full span with the student’s distribution when that span is replaced by a single surrogate embedding. This alignment is applied at every step and across many contexts, so any merge that systematically induces harmful drift in the local predictive distribution is penalized during training. Section 4.2 (Table 1) reports distribution-matching metrics and generation-level evidence(Table 8 reports qualitative greedy-decoding examples ) that merged spans preserve the base model’s outputs.

* **Regarding Mergering control and safety**

We clarified that UMIM can do precise ratio control (Input Stage): For the input prompt (prefilling), where the full sequence is available, we can precisely control the compression ratio with a ratio button. To address safety concerns in both prompting and decoding, we can employ a runtime "blacklist" to explicitly prevent the merging of specific named entities, idioms, or sensitive domain terms. This ensures that critical tokens remain separate and preserves fidelity in safety-critical applications.

* **Regarding comparison with some other works and positionl embedding**

We explained in detail the connections and differences between our paper and the papers mentioned by the reviewers, and clarified the positioning of our paper(detailed in rebutall). Regarding position encoding, we explained that the model's position encoding is assigned based on the key-value cache length. We do not need to make any special modifications to the position embedding.

* **Regarding the black box and N-grams size**

we explained our usage of black box  was intended to denote that the backbone model remains frozen and opaque to the optimization process. This is not a definition of a closed-source model; it only shows input and output. The choice of n-gram size conforms to language rules and is based on mathematical derivation and empirical analysis. Spans larger than 4-grams are uncommon in NLP, and merging 10-grams during decoding is slower than triggering 2-gram merging.

---

> ### Author Response · Authors · 2025-12-04
>
> * **Regarding Misunderstandings methods*
>
> About reviwer feel somewhat heuristc about our method. We made a detailed explanation.
> Our method has an mpirical evolution Our choice was not arbitrary.
>
> We used merge based on the cosine similarity of their static embeddings and so on. they doesn't work well.
> Our method aslo has a linguistic motivation: repairing tokenizer fragmentation Current subword tokenizers (like BPE) often fragment semantically unitary concepts due to vocabulary constraints. For example, the word "excitedly" might be split into ["excited", "ly"]. It also follows the training rule. high-frequency spans provide dense supervision signals across diverse contexts, allowing the merge module to learn a robust, context-agnostic representation that minimizes the teacher-student KL divergence.
>
> We also aim to address the common issues of excessive long sequence memory usage and slow speed in LLM. We implemented token merging at both the prefilling and decoding.
>
> "Finally, we wish to emphasize that our method is purpose-built to address existing real-world challenges. It is the result of rigorous empirical consideration and has been developed through iterative practical verification. We believe that our comprehensive experimental results, combined with the detailed exposition in the paper, provide compelling evidence of the superiority of our approach.  And resummarize our method, UMIM, a novel merge moudle that distills sequential Transformer computation into compact surrogate embeddings to accelerate inference and reduce sequence length without modifying or re-training the frozen base LLM. Crucially, our method goes beyond static prompt compression by uniquely integrating into both the pre-filling and autoregressive decoding phases: we introduce a dynamic "rollback" mechanism that detects and collapses generated token spans on-the-fly, actively reducing KV cache growth as generation proceeds.
>
> We are very grateful to AC and all the reviewers for their contributions to this year's ICLR. Due to the unique thing of this ICLR, we regret that we were unable to have more conversations with the reviewers. We express our respect to the reviewers who diligently reviewed the paper.

---

### Meta-Review · Area_Chair_xKqN · 2026-01-06

**Summary:**

The paper provides a technique aimed to reduce the computational complexity of transformers handling long input sequences. The authors provide a lightweight merge model allowing the main transformer model to handle a shorter sequence.

A major strength of the paper, according to the reviews, is the highly practical setting of the proposed solution, dnHy “The idea of compressing sequential computation post-training, without touching model internals, is interesting and practically useful..” 4jE3  “... operates only on input embeddings, requiring no access to model internals, no re-training, and no architectural changes … This makes it highly practical for deployment …”. It should be noted that some negative remarks were made about this aspect by fL57 ”The method does not truly treat language models as black boxes. The merge module must have access to …”. I read it not necessarily as a weakness of the method, but as a possible oversell in the paper (which can be easily fixed). It should be clear that the method is not black box, but it requires only a forward pass of the base model (required for the logits), rather than fine-tuning. Despite this comment, I see this as a strength of the paper.
Beyond the practical setup, reviews did mention positively the empirical evidence, both in terms of the bottom line performance: 4jE3 “The paper demonstrates strong empirical results. This up to 40% reduction in effective sequence length with minimal degradation in perplexity, downstream task accuracy (QA, summarization), and even math reasoning (AIME/AMC)”, and the extensiveness in terms of the backbone LLMs and task types: dnHy “Empirical study covers multiple backbones and task types, consistent results, and clear reporting of token reduction and perplexity trade-offs.”.

In terms of weaknesses, I saw two notable issues. The first is a comparison to baselines. Reviewers fL57 and dnHy mentioned baseline token merging techniques that should have been compared with the one presented here. Reviewing them, I found the baselines mentioned by fL57 incomparable as they refer to either vision or different (encoder-decoder) architectures. This being said, I think it is worthwhile adding some text in the paper explaining this difference, perhaps comparing the core ideas and explaining their differences. Reviewer dnHy mentioned zip2zip which seems to be a relevant baseline. The key difference is that it requires continued pre-training, putting it in a practical disadvantage. Nevertheless, the requirement to have a comparison with it does seem reasonable.

The second issue, which I perceive as the major one, is that of generalizability. This was raised by 3 reviewers (4jE3, KjpN, dnHy). They note that by training with a specific base model and its tokenizer, one one dataset (WikiText-103), it is unclear how the method will generalize across domains. The authors provided an experiment in the rebuttal where they show that when training the compression module on a different domain (coding), they get competitive results (superior to LLMLingua-2), but this doesn’t fully answer the question. One way to interpret the question of generalizability is whether the module trained in one domain will work on another (or how effectively will it work there), and that was not answered in the experiment. Second, the provided experiment is limited in scope (understandably since it is made in the rebuttal stage). This issue of generalizability seems to require more exploration. Given that it was raised by 3 reviewers and was not fully addressed in the rebuttal, I see this as a major weakness of the paper.

Concluding, with the main weakness pointed out by the reviews remaining, my recommendation is to follow the majority recommendation of rejecting the paper. I think it has potential, and the discussions are a great start towards strengthening the paper, but at this point it isn’t ready to be published.

**Reviewer Concerns:**

The concern of having a better comparison to baselines remain, though I do not see it as a major issue.
The concern of generalizability also remains, and this in my opinion, is a major issue.

**Reviewer Scores:**

Given that the concerns remain, I don't believe the reviewers would have changed their score.

---

### Decision · Program_Chairs · 2026-01-26

Reject